# Roles of Matrix Metalloproteinases and Their Natural Inhibitors in Metabolism: Insights into Health and Disease

**DOI:** 10.3390/ijms241310649

**Published:** 2023-06-26

**Authors:** Sébastien Molière, Amélie Jaulin, Catherine-Laure Tomasetto, Nassim Dali-Youcef

**Affiliations:** 1Institut de Génétique et de Biologie Moléculaire et Cellulaire Illkirch, 67400 Illkirch-Graffenstaden, France; sebastien.moliere@chru-strasbourg.fr (S.M.); jaulina@igbmc.fr (A.J.); cat@igbmc.fr (C.-L.T.); 2Centre National de la Recherche Scientifique, UMR 7104, 67400 Illkirch-Graffenstaden, France; 3Institut National de la Santé et de la Recherche Médicale, U1258, 67400 Illkirch-Graffenstaden, France; 4Faculté de Médecine, Université de Strasbourg, 67000 Strasbourg, France; 5Department of Radiology, Strasbourg University Hospital, Hôpital de Hautepierre, Avenue Molière, 67200 Strasbourg, France; 6Breast and Thyroid Imaging Unit, ICANS-Institut de Cancérologie Strasbourg Europe, 67200 Strasbourg, France; 7Laboratoire de Biochimie et Biologie Moléculaire, Pôle de Biologie, Hôpitaux Universitaires de Strasbourg, Nouvel Hôpital Civil, 67000 Strasbourg, France

**Keywords:** metabolism, MMP, TIMP, adipose tissue, obesity, insulin resistance, type 2 diabetes, NAFLD

## Abstract

Matrix metalloproteinases (MMPs) are a family of zinc-activated peptidases that can be classified into six major classes, including gelatinases, collagenases, stromelysins, matrilysins, membrane type metalloproteinases, and other unclassified MMPs. The activity of MMPs is regulated by natural inhibitors called tissue inhibitors of metalloproteinases (TIMPs). MMPs are involved in a wide range of biological processes, both in normal physiological conditions and pathological states. While some of these functions occur during development, others occur in postnatal life. Although the roles of several MMPs have been extensively studied in cancer and inflammation, their function in metabolism and metabolic diseases have only recently begun to be uncovered, particularly over the last two decades. This review aims to summarize the current knowledge regarding the metabolic roles of metalloproteinases in physiology, with a strong emphasis on adipose tissue homeostasis, and to highlight the consequences of impaired or exacerbated MMP actions in the development of metabolic disorders such as obesity, fatty liver disease, and type 2 diabetes.

## 1. Introduction

Matrix metalloproteinases (MMPs) are notoriously known calcium-dependent zinc-containing endopeptidase enzymes that degrade extracellular matrix (ECM), allowing tissue remodeling during development as well as in the postnatal life (reviewed in [1,2]).

The super-family of MMPs is composed of 26 members encoded by 24 different genes grouped into collagenases, gelatinases, stromelysins, matrilysins, membrane-type MMPs and others, based on their substrate specificity or cellular localization. The classification of MMPs, their structure (Figure 1), substrate specificity and the chromosome location of the genes encoding different MMPs are well represented in recent reviews [1,2,3].

The function of MMPs is balanced by their natural endogenous inhibitors or tissue inhibitors of metalloproteinases (TIMPs) [4,5]. The physiological roles of MMPs and TIMPs are numerous (for review, see Ref. [6]) and MMP deregulation can be seen in several pathological conditions ranging from cardiovascular diseases to fibrotic disorders, infections, neurological diseases, lung diseases, arthritis, inflammation, cancer etc.

Research on MMPs has expanded since the discovery of their involvement in cancer biology as modulators of the tumor microenvironment during tumorigenesis (reviewed in [7]). Research into the role of MMPs in metabolism has begun only recently and although many advances have been made in understanding the contribution of MMPs and their inhibitors to adipose tissue, lipid and glucose metabolism, some dark zones persist.

We aim to provide a comprehensive overview of the diverse roles of MMPs and TIMPs in metabolic functions and diseases in this review. Our primary focus will be on the intricate involvement of locally-active MMPs and TIMPs in adipose tissue (AT) and their dysregulation in obesity and associated metabolic dysfunction. Additionally, we will explore the contributions of MMPs and TIMPs in other key metabolic organs, including muscle, pancreas, and liver. By examining these aspects, we hope to shed light on the multifaceted impact of MMPs and TIMPs on metabolic homeostasis and disease pathogenesis.

## 2. Adipose Tissue Functions and Pathologies

In mammals, adipose tissue is central to the regulation of lipid and glucose metabolism. White adipocytes turnover and their metabolic profile are key determinants of whole-body fat storage and mobilization, and systemic insulin sensitivity. Alteration of adipocytes homeostasis may be seen in a variety of metabolic diseases, such as obesity, diabetes, cardiovascular diseases, as well as cancer-associated metabolic alterations.

Adipocyte differentiation and function are strongly regulated by their microenvironment and cross-talks with surrounding cells (other adipocytes, macrophage, endothelial cells etc.). Extracellular matrix of adipose tissue is composed of an isotropic matrix of a mix of collagen and elastic fibers: (i) cross-linked, thick bundles of type I collagen triple helix, highly resistant to general proteolysis, binding with fibronectin; (ii) type IV collagen forming the basement membrane; (iii) microfibers of type V and VI collagen linking type I fibers together with the basement membrane, thus regulating adipose tissue elastic resilience and expandability [8]. During the adipose tissue differentiation process, there is a shift from fibrillar collagen I, III and V-containing ECM in immature adipose tissue to basement membrane-associated collagen IV and VI in mature adipose tissue [9,10].

Adipose tissue enlargement following a high nutrient load requires a certain plasticity to cope with the increased amount of fat to be stored. Two complementary processes can take place [11,12]. Firstly, resident mature adipocytes increase their size through the uptake of fatty acids in a situation of energy excess, a process named adipocyte hypertrophy, which requires ECM remodeling. Simultaneously, adipose tissue can respond by increasing the number of adipocytes (hyperplasia) through enhanced recruitment and/or expansion of adipocyte precursors. Expression of basement membrane collagen type IV is increased by mesenchymal stem cells during adipogenesis and is remodeled by secreted MMPs during differentiation [13]. Type I, III, V and VI collagen expression levels are also increased during adipose tissue expansion [10].

Interestingly, in the context of obesity, adipose tissue plasticity differs between distinct kind of fat depots. Hence, hyperplasia was shown to be more pronounced in the subcutaneous fat of the lower body than in the upper body, whereas hypertrophy affects the subcutaneous fat of the upper body and the abdominal region more specifically, in response to overfeeding [12]. Accumulating evidence demonstrated that in conditions of excessive calorie consumption, the development of metabolic syndrome may be due to the absence or dysfunctional subcutaneous fat with states of insulin resistance, and the energy surplus will accumulate in visceral white adipose tissue (gonadal, perirenal, epicardial, retroperitoneal, omental or mesenteric) and organs that are not meant to store fat like the liver, skeletal muscle and the heart with deleterious consequences on health [14]. On the contrary, in individuals with insulin-sensitive subcutaneous adipose tissue, fat will accumulate in the latter preserving the other organs from undesirable fat deposition and these subjects, despite expansion of their subcutaneous fat, will be protected from metabolic syndrome development and its cardio-metabolic complications. Factors associated with preferential expansion of visceral fat and related insulin resistance include, among other causes, genetic susceptibility to visceral adiposity, smoking and maladaptive neuroendocrine response to stress [14].

MMPs are secreted in the ECM, or expressed at the surface of adipocytes, macrophages, endothelial cells, and other cell types. They play a major role in ECM homeostasis, adipocytes differentiation and growth, adipose tissue inflammation, insulin sensitivity, and, more generally, metabolic pathways [15]. One obvious effect of MMPs, which is ECM remodeling, directly influences cell growth and differentiation by modifying the microenvironment. Additionally, with the degradation of ECM components, some biologically active molecules such as fibroblast growth factor (FGF) or vascular endothelial growth factor (VEGF) can be released. In addition, many MMPs can cleave cell surface proteins and non-matrix substrates, cytokines or hormones, thus considerably increasing their range of action.

This section will review the known roles of MMPs and TIMPs in adipose tissue functions and diseases, with a strong focus on obesity and diabetes.

### 2.1. Gelatinases

The gelatinase family of MMPs consists of two members, MMP2 and MMP9 [3]. They have three repeats of a type II fibronectin domain in the catalytic domain, which binds to gelatin, collagens, and laminin [16]. Their name “gelatinase” comes from their ability to degrade denatured collagen/gelatin. MMP2, but not MMP9, can also cleave fibrillar collagen [17]. Both can also target non-fibrillar collagens type IV, V, VII, and X, as well as elastin and fibronectin [18].

While many studies determined the role of MMPs in inflammatory diseases and neoplastic disorders, there is increased evidence that MMP2 and MMP9 have an important role in adipose tissue homeostasis and its dysregulation.

#### 2.1.1. MMP2

MMP2, also known as gelatinase A, contributes to the remodeling ECM, especially in the normal bone, but also during development, wound healing, angiogenesis, and in disease, and tumor invasion.

MMP2 is initially secreted as inactive zymogen, known as pro-MMP 2, which is activated through the formation of a ternary complex with MMP14 and TIMP2 [19].

##### MMP2 Association with Obesity in Humans

Increased circulating MMP2 and MMP9 levels have been observed in obese subjects [20].

Interestingly, polymorphisms in the *MMP2* gene were associated with obesity in the Korean population [21]. Of the five *MMP2* variants analyzed, two synonymous single nucleotide polymorphisms (SNPs), namely rs17242319 (Gly226Gly) (now called rs1132896 in the SNP database BUILD 129) and rs10775332 (Phe602Phe) were significantly associated with overweight/obesity. In an independent study searching for sex-specific genetic associations of gene polymorphism with obesity and overweight [22], the frequency of the C/G genotype of the *MMP2* variant rs1132896 was higher in weight gainer women than in stable weight women compared to the common G/G genotype. Indeed, the C-allele was associated with a 2.5× higher risk of weight gain over a 10-year period in women as compared with carriers of the common G/G genotype [22]. No differences were observed in men regarding the *MMP2* variant, while polymorphisms in the *ACE* (angiotensin converting enzyme) and *FTO* (Fat mass and obesity-associated) genes were associated with weight gain. The role of estrogen has been suggested to explain this sex-selective association [23,24].

Lafontan and colleagues were the first to demonstrate that human adipose tissue can produce and secrete active MMP2 and MMP9, as evidenced by gelatin zymography experiment and qRT-PCR on mRNA extracted from human mature adipocytes. In this study, it was demonstrated that MMP2 and MMP9 expression increases during adipocyte differentiation, while their inhibition markedly reduced this differentiation, which suggests a potential important local role of MMP2 and MMP9 in adipose tissue homeostasis [25], representing a path toward the understanding of their role in obesity.

Type IV collagen, the main constituent of the basement membrane, is increasingly produced by the human mesenchymal stem cell during adipogenic differentiation. In parallel, the cells develop enzymatic capability to remodel the basement membrane, via activation of MMP2 and MMP9, with complex regulation by MT1-MMP (MMP14) and TIMP2, allowing remodeling that is required by the intracellular accumulation of fat [13].

##### MMP2 and Adipocyte Differentiation

Several experimental studies have tried to elucidate the mechanisms behind the role of MMP2 in adipocyte differentiation. Phenotyping genetically-engineered animals deficient in *MMP2*, challenged with either normal or high fat diet, brought new insights into the role of MMP2 in adipose tissue development and plasticity in response to nutrient load.

When challenged with a high fat diet (HFD) for 15 weeks, mice with genetic deficiency in *MMP2* had significantly lower body weights and accumulated less fat compared with littermates [26]. Importantly, *MMP2* deficient mice were significantly leaner than control animals before the onset of a HFD, and this was apparent as early as 5 weeks of age. In fact, *MMP2*^−/−^ animals have already been shown to have a lower body weight at birth due to a slower growth rate [27]. Similarly, wild-type mice treated with a gelatinase inhibitor (tolylsam) displayed a significant decrease in weight gain under a 15 week HFD regimen [26]. The reduced weight gain under a HFD in mice with either genetic inactivation or pharmacologic inhibition of MMP2 was mainly due to reduced fat pads and decreased adiposity (smaller adipocytes but greater adipocyte density), since neither feeding efficiency (weight gain normalized to caloric intake) nor energy expenditure (via measurement of spontaneous physical activity) were significantly altered. Interestingly, *MMP2*^−/−^ mice on a HFD had lower levels of cholesterol as compared to wild type mice, a feature that was not observed with tolylsam treatment [26]. In addition, *MMP2*^−/−^ mice had significantly decreased leptin levels.

Altogether, MMP2 seems to have an important role in adipose tissue development. Recent studies have brought some clues as to how the local pro-adipogenic effect of MMP2 is triggered [28,29].

Mouse embryonic fibroblasts (MEFs) obtained from *MMP2*^−/−^ mice showed impaired differentiation into mature adipocytes when subjected to an adipogenic differentiation medium, with a robust drop in intracellular lipid content and a significant reduction in pro-adipogenic markers such as peroxisome proliferator-activated receptor-γ (PPARγ) and its target adipocyte fatty acid binding protein (AP2) [28]. Genetic manipulation of *Mmp2* in 3T3-F442A pre-adipocytes confirmed these results: specific *MMP2* silencing using short hairpin-RNA knockdown reduced adipocyte differentiation, whereas specific *Mmp2* overexpression increased adipocyte differentiation, with a significant increase in the expression of PPARγ and AP2 at days 6 and 8 of differentiation [28].

In another experimental approach, using adipocytes xenograft, nude mice injected subcutaneously with 3T3-F442A cells in which *MMP2* expression was knocked down displayed unchanged fat pad formation, in terms of adipose tissue weight, adipocytes density and size, despite decreased adipogenic markers (AP2, PPARγ and adiponectin ADIPOQ) [28]. Such mixed results may be explained by the activity of the host’s own AT MMPs, and highlight the need for tissue-specific gene inactivation.

Other experiments allowed further insights on the time-dependent effect of MMP2 during adipogenesis. Treatment of 3T3-L1 pre-adipocyte with a specific MMP2 inhibitor hindered their differentiation into mature adipocytes dose-dependently, with an effect mainly in the early stages of differentiation, while little effect was observed in later stages. Mechanistically, the MMP inhibitor did not affect the expression pattern of the early inducer of adipogenesis, the CCAAT/enhancer-binding protein beta (C/EBPβ) transcription factor or its translocation to the nucleus, it dramatically reduced its DNA-binding capacity [30]. Chavey et al. also reported decreased differentiation of both 3T3L1-pre-adipocytes and primary rat pre-adipocytes into mature adipocytes in the presence of MMP inhibitor BB-94 (Batimastat) and showed that although the inhibitor did not affect mitotic clonal expansion, it did reduce expression of C/EBPβ [31].

Interestingly, the expression of the gatekeeper of adipogenesis preadiapocyte factor 1 (*Pref1*) was significantly increased during the early stage of differentiation and decreased in the later stages in 3T3-F442A cells overexpressing *MMP2*. This observation suggests an enhanced commitment of pre-adipocytes cells into the white adipocyte lineage in the early phase of differentiation, while the decreased *Pref1* expression in the late phase is synonymous with enhanced differentiation and adipogenesis. Indeed, *Pref1* is a marker of precursor cells and a factor whose expression is transiently increased in the early phase of differentiation and prevents adipogenesis, but its expression is known to drop drastically in the late phase to allow terminal differentiation [32]. Pref-1 is a known suppressor of adipogenesis though repression of the early adipogenic factors *C/EBPβ* and *C/EBPδ* through activation of an MEK/ERK/SOX9 pathway, preventing the activation of two important factors of terminal adipocyte differentiation C/EBPα and PPARγ. It remains to be determined how MMP2 directly or indirectly regulates PREF1 during the differentiation process.

##### MMP2 and Hypothalamic Leptin Signaling

MMP2 has been shown to intervene in genetic models of obesity, using two well-studied models of genetic obesity, specifically in two well-studied models involving ob/ob and db/db mice. These mice are genetically deficient in the gene responsible for producing the satiety hormone leptin and its receptor, respectively.

MMP2 expression levels have been reported to be robustly increased in epididymal fat pads of *ob/ob* and *db/db* mice [31]. It is firmly established that expression levels of leptin correlate with the mass of adipose tissue and that obese individuals express high circulating leptin levels (reviewed in [33,34]). Another report showed that treatment of *ob/ob* mice kept on a HFD for 15 weeks with a gelatinase inhibitor, Tolylsam, yielded lower body weight, weight gain, fat pad weight and fat blood vessel density but mild adipocyte hypertrophy, with unaffected food intake, body temperature and physical activity compared with untreated mice [35].

Mazor et al. have reported increased expression and activity of MMP2 in the hypothalamus of diet-induced obese mice [29]. Most importantly, the leptin receptor was demonstrated to be a substrate for MMP2, and MMP2 overexpression by a HFD was responsible for cleavage of the extracellular domain of the leptin receptor and subsequent impaired leptin signaling, a condition referred to as leptin resistance. Hypothalamic MMP2 inactivation restored leptin receptor expression and signaling, and significantly decreased weight gain in mice. Consistently, mice submitted to a HFD, bearing a mutation in leptin receptor that cannot be cleaved by MMP2, gained less weight than control mice. This effect of MMP2 on weight gain was shown to be mediated by NFκB activation [29]. This discovery has shed a new light after years of searching for the molecular mechanism behind leptin resistance since the identification of leptin by Friedman and colleagues in 1994 and highlights the complex multi-tissular roles of MMPs.

##### MMP2 and Type 2 Diabetes (T2D)

Increased expression levels of MMP2 and its inhibitors TIMP1 and TIMP2 were reported in the plasma of T2D patients, compared to healthy individuals [36].

Interestingly, a recent study highlighted the impairment of early steps of adipogenesis in type 2 diabetes, independently of obesity, by demonstrating an inverse correlation between the expression of *MMP2* and *AP2* in the stromal vascular fraction isolated from the abdominal subcutaneous adipose tissue of T2D obese patients vs. non-diabetic obese individuals [37]. In line with these results, it was shown that MMP2 (and MMP7) activities found in the sera change depending on the diabetes stage. Hence, it was observed that during decompensated T2D, the serum activity of MMP and the serum concentration of C-peptide—a marker of insulin secretion—decrease, while pro-insulin concentration robustly increases. On the other hand, the ratio of [proinsulin]/MMP activity was ~1:50 during the compensation phase, whereas in decompensated T2D this ratio was 1:12, suggesting that this ratio may become a diagnostic biomarker to assess diabetic decompensation [38].

#### 2.1.2. MMP9

MMP9, also known as gelatinase B, is produced by inflammatory cells and fibroblasts. Neutrophils, in particular, secrete MMP9, which plays a crucial role in ECM degradation and the subsequent release of proangiogenic factors. Additionally, infiltrating macrophages contribute to ECM remodeling by producing MMP9 [39].

MMP9 is initially synthesized as an inactive zymogen called pro-MMP9. Activation of pro-MMP9 occurs through the cleavage of its pro-domain by various proteases, including plasmin, urokinase-type plasminogen activator, and tissue-type plasminogen activator [39]. To maintain ECM homeostasis, MMP9 activity is tightly regulated by binding with tissue inhibitors of metalloproteinases (TIMPs).

##### MMP9 and Insulin Resistance

Unlike *Mmp2*, Mmp9-deficient mice did not differ from wild type littermates in terms of weight gain, adipocyte density and size when submitted to a high fat diet [40]. However, several human studies have linked MMP9 serum level with obesity-associated unhealthy metabolic state and insulin resistance.

MMP9 (together with VEGF-A) is upregulated in both subcutaneous and omental adipose tissue from morbidly obese patients, MMP9 expression level in fat tissue positively correlated with the homeostatic model assessment for the insulin resistance index [41]. Moreover, MMP9 expression in subcutaneous adipose tissue biopsies was shown to be positively correlated with body mass index and insulin resistance in impaired glucose tolerant subjects [42]. Treatment of patients with pioglitazone, a thiazolidinedione PPARγ agonist and insulin sensitizer, for 10 weeks dramatically reduced MMP9 expression. Pioglitazone also decreased MMP9 in 3T3-F442A adipocytes and THP1 macrophages. In addition, co-culture of adipocytes with macrophages enhanced MMP9 expression in adipocytes and pioglitazone decreased MMP9 expression in both cell types [42].

In the context of metabolic syndrome, serum concentration of MMP9 has been found to be positively correlated to age, systolic blood pressure, waist circumference and fasting blood glucose levels but negatively correlated to HDL cholesterol [43]. In another study, MMP9 and its inhibitors’ concentrations correlated with BMI, waist circumference, and metabolic parameters in patients with metabolic syndrome [44]. Moreover, higher concentrations of MMP9 were observed in metabolic syndrome patients with diabetes mellitus, compared to metabolic syndrome patients without diabetes [44].

To give strength to these data, a clinical study on patients with metabolic syndrome showed that when submitted to a regime with a high fiber/low-fat content and daily aerobic exercise for 3 weeks, their total level of serum MMP9 was significantly reduced along other markers of inflammation and oxidative stress as well as a significant decrease in circulating lipids, fasting blood glucose, insulin, and HOMA-IR index [45].

Importantly, Lee et al. found that the serum level of MMP9 did not change significantly 9 months after a bariatric surgery procedure aimed at reducing body weight (whereas MMP2 serum level decreased significantly), yet it inversely correlated with adiponectin, a well-established insulin-sensitizing adipokine, which supports the notion that MMP9 may be associated more with obesity-induced insulin resistance than with obesity itself [46].

Of note, in a mouse model of sucrose diet-induced insulin resistance, there was reduced AT expression and activity of MMP9 (and MMP2), with no change in MMP circulating levels nor in PPARγ expression [47]. This type of insulin resistance is different from high-fat diet-induced obesity, and insulin resistance has been ascribed to impairment of the hepatic parasympathetic nerves [48]. The role of MMP9 (and MMP2) in sucrose diet-induced insulin resistance needs to be studied in more detail.

##### MMP9 and “Metaflammation”

Treatment of human monocytic THP1 cells with the bacterial lipopeptide Pam3CSK4 was shown to induce the production of MMP9 and this effect was mediated by the TLR2/Myd88/JNK/p38MAPK/NFκB signaling pathway [49]. This pathway is notoriously known to mediate lipid and inflammasome-induced insulin resistance (reviewed in [50,51]). Hence, MMP9 could play a significant role in the maintenance of the state of chronic low-grade inflammation observed in obesity and commonly referred to as metabolic inflammation or “metaflammation”.

Recently, a study investigated the expression profile of markers of adipogenesis, inflammation and ECM remodeling in subcutaneous adipose tissue (ScWAT) biopsies and peripheral blood mononuclear cells (PBMCs) from overweight and obese subjects as compared to normal weight subjects [52]. Lower expression of adipogenesis markers, insulin signaling pathway and angiogenesis genes, and increased expression of ECM remodeling markers in ScWAT as well as enhanced expression of macrophage markers but not of other immune cells were observed in the overweight subjects. However, no difference in expression of proinflammatory markers in PBMCs was noted. In multiple regression analysis, the authors found that expression of *MMP9* along with *CEBPA*, *ADIPOQ*, *IRS1*, *IRS2* and *SLC2A4* was associated with insulin sensitivity independently of body mass index (BMI) [52].

Another study demonstrated that resident AT macrophages’ proliferation is induced first by nutritional overload, followed by recruitment of M1-like macrophages [53].

Together, these results point to the fact that macrophages are the first immune cells to infiltrate AT inducing ECM remodeling, which may be instrumental in insulin resistance development through the TLR2/Myd88/NFκB pathway [49].

### 2.2. Stromelysins

Canonical stromelysins, MMP3 and MMP10, are secreted as pro-enzymes and share similar ECM component substrates: fibronectin, proteoglycans, type IV collagens, and laminin. Unlike collagenases, they cannot degrade fibrillar collagen type I.

#### 2.2.1. MMP3

Beyond matrix proteins, MMP3 (stromelysin-1) can cleave a wide range of ECM molecules, among which plasminogen activator inhibitor-1, fibrinogen, plasminogen, as well as several pro-MMPs (such as pro-MMP1), marking its crucial role as the initial step in the MMP-mediated degradation and remodeling cascade [54].

MMP3 is secreted through exocytosis and extracellular vesicles by a variety of cell types (fibroblasts, macrophages, endothelial cells, epithelial cells, as well as cancer cells) and is a potent promoter of epithelial-to-mesenchymal transition [55].

MMP3 has been found to have important roles in adipose tissue physiology and inflammation.

##### MMP3 and Adipocyte Differentiation

*MMP3* is expressed in both subcutaneous (ScWAT) and gonadal (visceral) white adipose tissue (GWAT) in male mice under standard fat diet with higher expression levels in ScWAT as compared to GWAT. However, upon a HFD, GWAT expresses higher levels of *MMP3* as compared to ScWAT suggesting a depot-specific regulatory role of MMP3 in obesity [56].

MMP3 loss-of-function (LOF) studies point to an important regulatory role of MMP3 in adipose tissue growth. Lijnen and Coll. showed that male *MMP3*-deficient mice gain more weight and are hyperphagic when submitted to a HFD for a 15-week duration. Interestingly, at the end of the diet, analysis of fat pads showed that the weight of both ScWAT and GWAT fat depots were significantly higher in *MMP3*^−/−^ mice in comparison to controls. However, significant adipose tissue hypertrophy and increased adipose tissue angiogenesis was observed only in GWAT of *MMP3*^−/−^ mice but not in ScWAT, suggesting a preventive role of MMP3 from excessive visceral adipose tissue expansion in nutritionally induced obesity in male animals and that ScWAT probably expands through hyperplasia in the absence of MMP3 [57]. In addition, this study strongly supports that MMP3 may regulate food intake to limit excessive weight gain in mice.

In another study, major gender- and depot-dependent differences in *MMP3* (and *TIMP4*) expression were observed during adipose tissue remodeling when mice were challenged with a HFD. Wu et al. investigated the influence of a HFD on the size and number of adipocytes in male and female mice. In females, GWAT responds to a HFD with hypertrophy (increase in adipocyte size) and hyperplasia (increased fat cell number), whereas in males fat expansion occurs mainly through hypertrophy [58].

In vitro, overexpression of MMP3 in 3T3L1 pre-adipocytes or treatment of human pre-adipocytes with recombinant human MMP3 (rhMMP3) inhibited adipocyte differentiation, whereas concomitant treatment with recombinant TIMP4 prevented this effect and resulted in enhanced adipocyte differentiation [58]. Hence, MMP3 is regarded as an anti-adipogenic factor.

Adipocyte hypertrophy is hypothesized to trigger the recruitment of adipocyte progenitor cells (APCs) to allow adipose tissue capacity to expand when it reaches its maximal size. This capacity to recruit APCs is thought to be limited in obesity and may play an important role in metaflammation and its complications. *MMP3* is mainly expressed by APCs [58], whereas its inhibitor *TIMP4* is essentially expressed by mature adipocytes [56] (*TIMP1*, *TIMP2* and *TIMP3* are mainly expressed in the stromal vascular fraction).

Fat depot *MMP3* expression and the abundance of APCs is not similar in males and females. Under regular diet, MMP3 expression is higher in ScWAT compared to GWAT, suggesting that Mmp3 may exert a higher adipogenic inhibitory effect in ScWAT than in GWAT, and this differential expression is more pronounced in female mice than in males. However, the expression of *MMP3* is higher in GWAT in females compared to males, suggesting that Mmp3 may regulate fat depot distribution differently in males and females. Under regular diet, the abundance of APCs in GWAT and ScWAT is higher in females compared to males, consistent with the higher *MMP3* expression in female GWAT and ScWAT compared to males. Under a HFD, a significant increase in *MMP3* expression is observed in all fat depots (GWAT, mesenteric WAT and ScWAT) in males whereas a modest yet significant increase is found only in GWAT in females [58]. In addition, under a HFD the abundance of APCs and the expression of *MMP3* in APCs is increased in GWAT in males but not in females. The higher Mmp3 expression in GWAT in males restrains in an autocrine loop the progression of APC differentiation into adipocytes, allowing the existing adipocytes to grow further and become dysfunctional or die. Inflammation will ensue through activation of macrophages, which contribute to the decrease in adipogenesis in a paracrine loop. In female GWAT, HFD induces a decrease in MMP3MMP3 protein levels that will allow an increase in adipocyte number through APC’s differentiation, preserving thereby the functionality of the adipose tissue, which will expand both by hypertrophy and hyperplasia. Interestingly, the higher protein *MMP3* expression in female ScWAT is decreased under a HFD, allowing the differentiation of preadipocytes and expansion of ScWAT by hyperplasia. In male ScWAT, HFD feeding induces *MMP3* expression in preadipocytes impeding their differentiation and AT grows by hypertrophy of existing adipocytes. These differences in the abundance of APCs and APCs’ Mmp3 expression may explain, at least partly, the sex-, fat depot- and diet-dependent differences to regulate hyperplastic versus hypertrophic adipose tissue expansion [58].

Other factors contribute to the aforementioned differences. For instance, a HFD induces a dramatic increase in the expression of the proinflammatory macrophage marker *Cd11c* in adipose tissue of males in comparison to female fat depots where the increase was only modest [58], suggesting a sex dimorphism in inflammatory markers under nutritional challenge. In addition, significant sex- and fat-dependent differences in *Cd11c* expression are noted under a regular diet, although they are much less pronounced. Hence, *Cd11c* expression is more expressed in GWAT than in ScWAT in males, whereas in females *Cd11c* is expressed more in ScWAT than in GWAT (this profile is inversed under a HFD), with a higher expression in males than in females [58]. Other factors are involved in this process, such as the Mmp3 modulator TIMP4, which is also regulated in a gender- and depot-dependent manner. TIMP4 will be discussed in more detail below in a dedicated section.

Altogether, these observations show that significant sex- and depot-dependent differences in Mmp3 and Cd11c expression exist during adipose tissue remodeling and that modulation of the Mmp3/TIMP4 ratio regulates this process.

*MMP3* is strongly expressed during mammary gland involution stimulated by weaning and *MMP3*^−/−^ mice display precocious differentiation and hypertrophy of adipocytes in the mammary gland environment, whereas epithelial cell death is unaltered [59]. Interestingly, in vitro *MMP3* along other MMPs (*MMP14*, *MMP13* and *Mmp2*) and TIMPs (*TIMP1-3*) are induced in committed pre-adipocytes, but only differentiated adipocytes express an activated and secreted MMP2 observed by increased gelatinase activity as analyzed by gelatin zymography. Hence, Mmp3 secreted by adipocytes may be regarded as a feedback negative regulator and a guardian of the adipocyte differentiation rate, limiting thereby excessive differentiation during mammary gland involution [59].

Interestingly, in a recent nutritionally-induced obesity murine model, it has been shown that phenol-rich water extract from purple maize pericarp (PPE), known to modulate inflammatory and obesity markers, significantly reduced body weight gain and markedly upregulated the expression of *MMP3* in the adipose tissue of HFD-fed mice [60].

Few human studies have been conducted on the metabolic role of Mmp3, yet a polymorphism in the *MMP3* gene (Lys45Glu or G allele carriers) was associated with a higher risk of severe obesity compared with A/A genotype carriers in a Tunisian population [61]. In this study, variants in *MMP7* (-181G) and *MMP12* (-82G) were also associated with obesity and its severity.

##### Pro-Inflammatory Effect of MMP3

A deleterious pro-inflammatory effect of MMP3 was previously reported, as MMP3 was shown to be secreted by adipose tissue-infiltrated macrophages. Hence, MMP3 seems to play a role in fatty-acid induced insulin resistance and angiogenesis through regulation of tumor necrosis factor α (TNFα) and the proangiogenic factor vascular endothelial growth factor (VEGF), respectively.

First, it was demonstrated that implantation of 3T3-L1 pre-adipocytes in visceral fat robustly up-regulates *MMP3*, and leads to an increase in TNFα secretion [62]. Most interestingly, it was shown that a HFD promotes certain adipocytes to co-express toll-like receptor 2 (TLR2) and its downstream molecule, TNFα to induce insulin resistance in mice in the presence of free fatty acids (FFA) [63]. Using macrophages-conditioned medium [64] and macrophage-3T3-L1 co-cultures [65], it was shown that MMP3 secreted from macrophages stimulates FFA-induced expression of VEGF by the adipocytes, via toll-like receptor 2 (TLR2) expression and activation.

The paracrine crosstalk between macrophages and adipocytes is notoriously known to promote insulin resistance mediated by metaflammation [51,66]. MMP3 appears to contribute to the multiple molecular aspects of insulin resistance pathogenesis in obesity through inflammation since treatment with an MMP3 inhibitor abolished MMP3-mediated secretion of TNFα.

Altogether, MMP3 seems to play a dual role in adipose tissue biology; it negatively controls adipose tissue expansion in a sex- and depot-dependent manner, while MMP3 overexpression in visceral fat favors inflammation, insulin resistance and angiogenesis. Thus, fine-tuning MMP3 expression in adipose tissue appears to be important to ensure proper adipogenesis and prevent insulin resistance.

#### 2.2.2. MMP10

MMP10 is expressed in epithelial cells, it allows keratinocytes to migrate during wound healing, by cleaving several components of hemidesmosomes, desmosomes, and associated cell adhesion molecules, as well as soluble bioactive and proMMP9 [55,67].

MMP10/stromelysin-2 does not appear to contribute significantly and directly to adipose tissue development and angiogenesis in a model of dietary-induced obesity [68]. However, microarray data identified MMP10 among the genes that were highly upregulated in human adipocytes in response to macrophage-conditioned medium for 4 h and 24 h. MMP10, MMP1, MMP3, MMP9, MMP12 and MMP19 were also robustly induced, with MMP1 and MMP3 being the most secreted proteins by adipocytes [69].

#### 2.2.3. MMP11

The third member of the stromelysin family, stromelysin-3/MMP11, is distantly related to other stromelysins. It is secreted as an active protease by fibroblasts, it can cleave collagen VI and non-structural substrates such as α-1 antitrypsin, α-1 proteinase inhibitor and IGFBP-1, but is not able to degrade major ECM components [55]. Increased MMP11 expression in the tumor microenvironment (cancer-associated adipocytes and cancer-associated macrophages) was documented in several invasive cancers, including breast cancer [70,71,72]. MMP11 was initially identified in our laboratory back in 1990 as a factor secreted by stromal cells of breast carcinomas [73]. Since then, several discoveries have shed light on the role of MMP11 in metabolism.

##### MMP11 Protects against Obesity-Associated Metabolic Dysfunction

One study initially identified an association between MMP11 and adipogenesis, since HFD-fed *MMP11* deficient mice had adipocyte hypertrophy [74]. Later, MMP11 deficiency was associated with increased adipose tissue expression of adipocyte differentiation master genes *Pparg* and *Fabp4* (*ap2*) [75]. In this study, the direct role of MMP11 in adipogenesis was uncovered. Hence, *MMP11*^−/−^ MEFs were more prone to be differentiated in adipocytes and treatment of preadipocytes with recombinant MMP11 delayed adipocyte differentiation, demonstrating a negative regulatory function of MMP11 on adipogenesis [75].

Next, our group generated a complementary *Mmp11* gain of function mouse model (*MMP11*-Tg, GOF) in which MMP11 is expressed in the whole animal through an epithelial specific promoter [76]. By studying *MMP11*-Tg mice and *MMP11*^−/−^ (LOF) animals in comparison to their respective controls, we brought evidence that MMP11 is an important regulator of whole-body energy homeostasis. Hence, *MMP11* overexpressing mice were lean, had improved metabolic parameters and were protected from diet-induced obesity; whereas *MMP11*^−/−^ displayed a phenotype reminiscent of the metabolic syndrome [76]. Interestingly, despite the beneficial effect of MMP11 on weight loss, *MMP11*-Tg animals exhibited mitochondrial dysfunction in the brown adipose tissue due to increased oxidative stress and a switch in metabolism from oxidative phosphorylation to aerobic glycolysis. This phenotype was probably mediated by an increase in the IGF1/AKT/FOXO1 signaling pathway, since *MMP11*-Tg mice had increased serum IGF1 concentration and activation of the IGF1/IGF1R/AKT cascade [76]. Notably, IGFBP1, the protein that controls the levels of circulating IGF1, was previously identified as a substrate for MMP11 [77] and our group has recently confirmed that recombinant MMP11 cleaves IGFBP1 in vitro (Tomasetto and Coll., 2022, unpublished data).

##### MMP11 and Metabolic Reprogramming in Cancer

In a follow-up study, an effort was put to unravel the role of such MMP11-driven metabolic alterations in the specific context of breast cancer and tumor progression. For this purpose, we crossed *MMP11*-GOF and LOF mice with animals expressing the Mouse Mammary Tumor Virus (MMTV)-polyomavirus Middle T-antigen (PyMT), a mouse model of breast cancer [78]. This study showed that MMP11 favors early mouse mammary gland tumor development through increased proliferation and reduced apoptosis. Mechanistically, in tumors from *PyMT^Tg^: MMP11^Tg^* mice, MMP11 activates the IGF1/AKT/FoxO1 signaling pathway, promotes endoplasmic reticulum stress response (UPR^ER^) and alters mitochondrial unfolded protein response (UPR^mt^) as compared to *PyMT^Tg^: MMP11^WT^* control mice. MMP11 also changes the metabolic profile of *PyMT^Tg^: MMP11^Tg^* tumors, as it promotes lipid intake and utilization while favoring a metabolic reprogramming from oxidative phosphorylation to aerobic glycolysis [78].

Altogether, MMP11-mediated alterations provide tumors with a significant advantage for growth and survival. Pharmacological inhibition of MMP11 should be a serious objective to achieve slow tumor growth in the early phase of its development, as it targets multiple processes involved in tumorigenesis. Reciprocally, IGF-1 was shown to promote MMP11 expression in gastric cancer cells, in a dose-dependent manner, through a JAK1/STAT3 signaling cascade to increase their proliferation and invasion [79].

##### Role of MMP11 in TNFα-Induced Insulin Resistance

Paradoxically, from a purely metabolic point of view, a recent study revealed that MMP11 is overexpressed in TNFα-induced insulin resistance in differentiated cultured 3T3-L1 adipocytes and in WAT of obese C57BL/6J mice compared to lean mice. The authors suggest that dysregulated MMP11 occurs early in adipose tissue dysfunction, leading to obesity and subsequent insulin resistance [80]. The question that arises from these observations is the following: does MMP11 protect from insulin resistance, or is it a deleterious factor? Our data support the idea that “too much of a good thing, harms”, as in MMP11 expression is necessary for glucose homeostasis, since its absence promotes insulin resistance in mice, whereas its overexpression in epithelial cells (Keratin-14 promoter used in *MMP11*-Tg mice) improves glucose tolerance and insulin sensitivity, although it alters mitochondria function in adipose tissue to a certain extent through increased oxidative stress. It is also plausible that in response to a metabolic insult such as nutrient stress-induced insulin resistance, the organism responds by increasing MMP11 expression, perhaps in an attempt to restore metabolic homeostasis. Adipose tissue-specific overexpression of MMP11 will certainly bring more insight onto the complex role of this protease in energy metabolism.

##### Expression of MMP11 during Adipocytic Differentiation

Regarding adipocyte differentiation, our group has shown that like *MMP2*, *MMP11* expression seems to be regulated differently during the early and late phase of differentiation but in an opposite manner. In days 1–2 of MEF differentiation, *MMP11* expression is robustly decreased, then it is upregulated starting day 4 concomitantly with the increased expression of the adipogenic marker *Fabp4*/*ap2*, to culminate at day 6 of adipocyte differentiation [75]. Hence, MMP11 seems to play a dual role during adipocyte differentiation.

Consistently, in another study *MMP11* expression was shown to be significantly increased at d4 of differentiation and decreased at a late stage (d12) of differentiation, while *MMP11* overexpression decreased fat accumulation [81]. Interestingly, this report identified the non-coding RNA miR125b-5p as a direct negative regulator of MMP11 during adipogenesis with a robust increased expression of *miR125b-5p* in late adipogenic differentiation [81]. By contrast, artificial overexpression of *miR125b-5p* blocks adipogenesis consistent with a decrease in expression of *Cebpb* (encoding C/EBPβ), and *Cebpd* (encoding C/EBPδ), through a mechanism yet to be identified [81]. The exact role of MMP11 on the expression of pre-adipocyte markers such as Pref1 and the commitment of pre-adipocytes into the adipocyte lineage are still undetermined. In addition, the relevance of the miR125b-5p/MMP11 interaction in the context of breast cancer biology is yet to be explored.

### 2.3. Collagenases

Collagenases are capable of cleaving fibrillar collagen in the triple-helical domain, leaving the molecule unstable, in a gelatin conformation that can then be further degraded by other MMPs [82].

#### 2.3.1. MMP1

The first vertebrate collagenase to be purified and cloned, MMP1, also known as collagenase-1 or fibroblast collagenase, is the prototype for other collagenases.

MMP1 exhibits a remarkable capability to interact with and unwind the triple-helical domain of collagen. This interaction involves a conformational change that encompasses the catalytic and hemopexin-like domains of MMP1. As a result, a collagen triple helix is exposed, facilitating its subsequent digestion by other MMPs [83]. Besides its role in collagen degradation, MMP-1 also possesses the ability to degrade other components of the extracellular matrix (such as perlecan and aggrecan) as well as non-matrix proteins (including insulin-like growth factor binding proteins (IGFBPs) type 3 and 5, tumor necrosis factor-α, antichymotrypsin, and antitrypsin). These diverse functions make MMP-1 a multifunctional protein [84].

In the context of tissue remodeling, MMP1 plays a critical role in facilitating the breakdown and turnover of the extracellular matrix. This is particularly significant in adipose tissue, where the excessive accumulation of collagen type I is indicative of unhealthy adipose tissue. Therefore, MMP1’s involvement in ECM remodeling is of utmost importance, helping to maintain tissue homeostasis and proper functioning.

##### MMP1 Polymorphism and Metabolic Diseases

Genetic variants in the *MMP1* gene were shown to be associated with obesity traits in the Korean, Taiwanese and Chinese populations [85,86,87]. On the other hand, MMP1 serum levels have been shown to be significantly increased in patients with type 2 diabetes [88]. Interestingly, diabetic patients with cardiovascular heart disease (CHD) had higher frequency of a genetic polymorphism in the promoter region of *MMP1* (2G/2G) and a higher MMP1 plasma level [88]. These findings were recently confirmed in a Han Chinese population [89]: the frequency of the 2G allele of *MMP1* (variant rs1799750) was enhanced in diabetic patients with CHD, and increased BMI was also correlated with the 2G/2G genotype, suggesting that this polymorphism may favor the development of CHD in overweighed/diabetic patients [89].

Moreover, *MMP1* was recently identified using bioinformatics analysis as a hub gene in T2D when searching for differentially expressed genes and molecular pathways in pancreatic islets from T2D patients as compared to normal islets obtained from three different gene expression public datasets [90].

##### MMP1 Secretion during In Vitro Adipocytes’ Differentiation

Macrophage-mediated robust overexpression and release of MMP1 (and MMP3) by pre-adipocytes is mediated by IL-1β through the MAPK signaling pathway [91]. *MMP1* (and *MMP3*) are expressed by pre-adipocytes, then robustly downregulated after differentiation [92,93]. The interaction between infiltrating macrophages and pre-adipocytes to promote MMP secretion and ECM remodeling is probably a key factor during AT expansion.

##### MMP1 Decrease and Collagen I Accumulation in Murine Obese Adipose Tissue

During a prolonged HFD, C56BL/6J mice have increased adipose tissue levels of collagen I (*Col1a1* and *Col1a2*) and *TIMP1*, and lower levels of *MMP1* [94], consistent with the role of MMP1 in regulating collagen I deposition, a marker of fibrosis associated with insulin resistance, during dietary and genetic obesity [95,96].

Interestingly, a significant elevation in Toll-like receptor-2 (TLR2) expression, a sensor of saturated fatty acids that elicit inflammatory responses promoting insulin resistance (see review [51]), was also observed under a HFD, while *Tlr2*^−/−^ mice fed with a HFD had decreased levels of *Col1a1*, *Col1a2*, and *TIMP1* but increased *MMP1* transcripts compared with WT fed the same diet [94]. Downstream effectors of TLR2, namely MYD88 and the stress-activated p38 MAPK, were significantly increased in obese mice, as well as pro-inflammatory cytokines (*Tnfa* and *Il6*) and were robustly reduced in *Tlr2*^−/−^ mice under a HFD compared to WT under the same diet [94]. Altogether, this study demonstrates that TLR2 mediates collagen accumulation and pro-inflammatory cytokines’ activation in WAT during diet induced obesity, involving *MMP1* and *TIMP1* imbalanced expression.

#### 2.3.2. MMP8

MMP8, the second member of the collagenase family, also known as collagenase-2 or neutrophil collagenase, is well-known to be secreted by neutrophil cells during inflammation [97,98]. MMP8 cleaves collagen (Type I, III, II, VII, X) and a diverse array of other non-collagenous ECM substrates (e.g., fibronectin, tenascin, laminin-5, proteoglycans, cartilage aggrecan, fibrinogen, etc.) and non-structural substrates including cytokines, serine protease inhibitors, signaling molecules, growth factors (e.g., hepatocyte growth factor or HGF) and other molecules such as angiotensin, IGFBP5 and L-selectin (for details see review in [97]).

In normal healthy tissues, MMP8 expression is generally minimal or absent, as it is primarily associated with acute and chronic inflammation [97], such as in cancer, vascular diseases, innate immune response or metabolic diseases.

Thus, significant elevated circulating MMP8 levels and MMP8/TIMP1 ratio were documented in obese young individuals compared with normal weight subjects in a Finnish population, and smoking was reported to increase further MMP8 levels and MMP8/TIMP1 ratio in both groups [99]. Moreover, increased plasma concentrations of MMP8 and TIMP1 (along other pro-inflammatory cytokines) have been previously reported in patients with features of the metabolic syndrome (MeS) as compared with healthy controls [100]. Another study reported increased circulating MMP8 (along MMP9 and TNFα) levels in diabetic patients with MeS compared with non-diabetic patients with MeS in a Tunisian cohort of patients [101]. These observations should be confirmed in preclinical models of diabesity using MMP8 specific inhibitors to see whether these molecules can mitigate/reverse insulin resistance. Interestingly, early decrease of circulating MMP8 levels was recently demonstrated to correlate with leptin reduction and predict diabetes remission in patients that underwent bariatric surgery at a 3-year follow-up timepoint [102]. This study gives an additional argument in favor of MMP8 inhibition to potentially decrease/reverse insulin resistance and diabetes in obese patients.

Mechanistically, an in vitro study reported that human insulin receptor can be cleaved by MMP8 and this observation could explain, at least partly, the role of MMP8 in impaired insulin signaling and insulin resistance in obese individuals [99]. Consistently, salivary MMP8 (and MMP9) levels were shown to correlate with HbA1c in non-obese type 1 diabetic patients [103], suggesting that these non-invasive salivary markers may predict glycemic control in these patients. This hypothesis needs to be validated in a large cohort of T1D patients and investigated in T2D patients.

Finally, increased MMP8 serum levels have been associated with gestational diabetes mellitus [104]. MMP8 concentration at baseline was recently shown to correlate weakly with pre-pregnancy BMI and was significantly associated with late, but not total gestational weight gain [105]. In this study, an inverse correlation of MMP8 concentration with birth weight was also documented, and this association was not affected by pre-pregnancy BMI [105]. It was suggested that MMP8-mediated inflammation of the chorion may be responsible for the lower birth weight as MMP8 was previously associated with chorioamnionitis [106] and premature term delivery [107]. The causal role of MMP8, if any, in this context needs to be further explored.

#### 2.3.3. MMP13

MMP13, also known as collagenase-3, exhibits a broader substrate range and generates diverse degradation products, highlighting distinct collagen-cleavage mechanisms [108]. It effectively cleaves collagen I, the primary collagen in bone, as well as collagen II fibrils, which are crucial in cartilage. The role of MMP13 in ECM remodeling has been investigated in various contexts, including fetal bone development, post-natal bone remodeling, and gingival homeostasis.

MMP13 is primarily secreted by fibroblasts, chondrocytes, and osteoblasts [108].

Interestingly, *MMP13* is expressed at a relatively low level in pre-adipocytes and is highly expressed in fully differentiated adipocytes, underscoring an important role in adipocyte differentiation [109]. *MMP13* expression is relatively higher in ScWAT than GWAT under standard diet, while this profile is inverted under a HFD with an approximately four-fold expression in GWAT compared to ScWAT [109], suggesting a contribution of MMP13 in the pathogenesis of obesity.

Interestingly, *MMP13* is more expressed in the stromal vascular fraction (SVF) than in adipocytes and this expression is exacerbated in the SVF in both GWAT and ScWAT under a HFD [109], implying the role of MMP13 in adipose tissue and ECM remodeling during obesity. Moreover, *MMP13* mRNA expression levels are correlated with the degree of obesity in GWAT but not in ScWAT [109], underscoring the prominent role of MMP13 in the development of visceral obesity. Consistently, selective inhibition of MMP13 with the CP-544439 compound suppressed adipocyte differentiation of 3T3-L1 cells. In vivo, CP-544439 reduced fat mass, adipocyte size and increased insulin sensitivity in mice fed a HFD [110]. Hence, pharmacological inhibition of MMP13 may prove efficient against obesity and its complications.

### 2.4. Elastase

Among the matrix metalloproteinases (MMPs) able to degrade elastin, MMP12 exhibits high specificity for elastin degradation. It is primarily secreted by stimulated macrophages, playing a key role in inducing elastin degradation [111]. MMP12 expression is observed in placental tissue in both macrophages and stromal cells, but it is not typically expressed in normal adult tissue [112].

Concerning the potential role of MMP12 on metabolic functions, phenotype studies of *MMP12*^−/−^ mice have been somewhat conflicting. Lijnen et al. showed that MMP12 deficiency had no direct impact on adipose tissue mass (both ScWAT and GWAT), cell density and overall total body weight in mice compared to wild type mice both on a chow diet and a HFD, yet a significantly lower amount of crown-like structures (CLS) (activated macrophages, mostly derived from monocytes, surrounding dead or dying adipocytes and are hallmarks of a proinflammatory process in adipose tissue) was observed [113]. In sharp opposition, Heinecke et al. showed that *MMP12*^−/−^ mice fed with a HFD displayed higher body weight, increased fat mass and adipocyte size but lower lean mass, accumulated more M2c macrophages (CD14^+^/CD206^+^/Cd11c^−^ cells) in their adipose tissue and were more insulin sensitive. It is worth mentioning that the M2c population of macrophages are known to be induced by glucocorticoids and are referred to as anti-inflammatory, reparative, “deactivated” macrophages, different from the typical “alternatively activated” M2a macrophages (CD14^low^/CD206^+^/Cd11c^−^) induced by IL4 and IL13 (for review see ref [114]). Most interestingly, CD206 (C-type mannose receptor 1) protein was shown to be induced during inflammation to promote the clearance of inflammatory proteins and *Cd206* deficiency (*Mr1^−/−^*) yielded significantly increased circulating inflammatory markers in mice [115]. Moreover, expression of nitric oxide synthase 2 (Nos2), a marker of M1, classically activated, proinflammatory macrophages and insulin resistance, was markedly decreased in adipose tissue of *MMP12*^−/−^ mice (but also in stimulated dendritic cells and macrophages), while significantly increased in WAT of *MMP12*^+/+^ mice under a HFD, suggesting that MMP12 deficiency alters macrophage recruitment and polarization under nutrient stress in mice. These results suggest that MMP12, secreted by macrophages and dendritic cells (DC), promotes insulin resistance mediated by metaflammation, while restraining adipose tissue expansion in mice [116]. While *MMP12* expression does not correlate with body mass index in human subjects, it correlates positively with M2c macrophages density in ScWAT, HOMA-IR and inflammatory mediators such as adipose tissue *TNFA* and *CCL2* mRNA expression levels and inversely with *ADIPOQ* transcripts in obese individuals [116]. It is yet to be = determined whether the same changes occur in visceral fat of obese individuals. In mice, *MMP12* transcripts are dramatically induced in obese adipose tissue compared to lean tissue [31,56]. Of note, DCs and M2c macrophages express much higher levels of *MMP12* than M1 macrophages. Altogether, data from *MMP12*^−/−^ mice and from obese individuals converge to the notion that increased M2c macrophage-expressing *MMP12* levels in adipose tissue of obese subjects may be a compensatory mechanism to restrain adipose tissue expansion and limit inflammatory mediators to counteract insulin resistance.

Recently, the role of MMP12 has been investigated in microbiota-dependent metabolic alterations and insulin resistance in mice and obese patients. Indeed, it was demonstrated that MMP12^+^ macrophages link microbiota-dependent inflammation and mitochondrial OXPHOS impairment in WAT of high fat-high sucrose (HFHS)-induced T2D mice compared to non-diabetic mice under a normal diet [117]. Consistently, *MMP12*^−/−^ conventional mice, but not germ-free *MMP12*^−/−^ animals, exhibit improved insulin resistance compared to WT mice, confirming the important role of MMP12 in mediating intestinal microbiota-induced metabolic alterations in T2D. Interestingly, MMP12 treatment of adipocytes provokes insulin resistance, whereas treatment with the MMP12 inhibitor MMP408 reduces fasting glycemia and improves glucose tolerance in HFHS diabetic mice in the presence of microbiota. In humans, MMP12^+^ macrophages signature in the WAT is also associated with insulin resistance in obese subjects [117]. This study showed that HFHS-induced MMP12 in the adipose tissue bridges microbiota-dependent adipose tissue inflammation and insulin resistance. MMP12 was shown to be induced by HFHS in macrophages through a MYD88-ATF3–dependent pathway, yet more efforts are needed to deepen our understanding of the molecular mechanisms underlying MMP12-mediated insulin resistance. From a therapeutic perspective, MMP12 emerges as a promising target to resolve T2D-mediated metabolic abnormalities.

### 2.5. Matrilysins

MMP7 and MMP26 are structurally distinct from other MMPs in that they lack the hinge region and hemopexin domain, which is required for interactions with other MMPs and TIMP.

#### 2.5.1. MMP7

MMP7 digests gelatin, but has no action on collagen I, II, IV, V. MMP7 is expressed in epithelial cells, including in the endometrium, exocrine glands of skin, salivary glands, pancreas, lung, bile ducts, and breast [118].

MMP7 plays a crucial role in epithelial repair and proliferation by being recruited in cholesterol-rich domains of the plasma membrane, (bounded MMP7 is active and resistant to TIMP inhibition) and therefore promoting membrane-associated growth factors [119]. In the human endometrium, MMP7 expression increases during menstruation and the proliferative phase, contributing to endometrial regeneration [118].

Expression levels of *MMP7* in obesity is controversial and contradictory results were reported in a number of studies. First, a positive correlation was reported between serum MMP7 concentration (but also of MMP1, -3, -8, -9 and -12) and BMI in 54 healthy reproductive-aged women and significantly higher MMP7 concentrations were found in women with higher BMI as compared to women with normal BMI [120]. Second, *MMP7* expression levels (and of *COL6A3* encoding collagen VI alpha) were found to be increased in ScWAT of obese insulin resistant subjects as compared with obese-insulin sensitive subjects, suggesting an involvement of MMP7 in obesity, ECM remodeling and insulin resistance [121].

However, in two animal models of genetic obesity (*ob/ob* and *db/db*) and in a HFD model, while mRNA levels for *MMP2*, *MMP3*, *MMP12*, *MMP14*, *MMP19*, and *TIMP1* are robustly induced, *MMP7* (and *MMP3*) transcripts are markedly reduced in obese visceral adipose tissue compared to lean tissue [31]. This observation was corroborated by the study from Maquoi et al., where a significant decrease in *MMP7* transcripts was demonstrated in adipose tissue (GWAT and ScWAT) from obese mice (DIO) compared to adipose tissue from mice under a standard diet [56]. Interestingly, in gonadal fat, *MMP7* is expressed mainly in SVF with marginal expression in adipocytes whereas in ScWAT, *MMP7* is expressed at higher levels in adipocytes [56]. Moreover, a negative correlation is found between *MMP7* expression and gonadal adipose tissue weight, but not in ScWAT, suggesting the different role of MMP7 in GWAT and ScWAT [56]. More efforts are needed to understand the distinct role of MMP7 in adipose tissue subpopulations of cells both in visceral and subcutaneous WAT and investigate the consequences of deregulated MMP7 in diabesity. Furthermore, in a human study it was shown that weight loss by bariatric surgery was accompanied by a significant increase in circulating levels of MMP7, though without reaching those of lean subjects [122], consistent with preclinical data mentioned above [56].

Recently, while investigating the role of gut microbiota dysbiosis in metaflammation and obesity development, it was demonstrated that neurotensin (NT), an endocrine hormone secreted by enteroendocrine N-cells of the small intestine in response to fat ingestion [123,124,125], is a significant contributor to gut microbiota dysbiosis and inflammation after high fat feeding through disruption of the MMP7/α-defensin (DEFA) axis. Indeed, NT deficiency prevents decreased MMP7/DEFA5 expression and abnormal bacteria composition in the intestine upon a HFD [126]. A previous seminal work from the same group unraveled a direct link between neurotensin and obesity that is conserved across species [127]. This study remarkably shows that NT deficient mice exhibit significant reduction in intestinal fat absorption, are protected from obesity, hepatic steatosis, and insulin resistance. Moreover, NT expression in drosophila promotes fat accumulation in midgut, fat body and oenocytes (hepatocyte-like cells), and inhibits AMPK activation. This mechanism seems to be conserved across species, as human obese and insulin resistant individuals have elevated circulating levels of pro-NT, confirming a dysregulation of NT expression during nutrient-induced obesity [127]. Moreover, higher circulating pro-NT levels were recently associated with increased risk of nonalcoholic fatty liver disease (NAFLD) and a predictor of NAFLD independently of other metabolic risk factors [128]. Finally, the NT variant rs2234762 C > G was recently identified to be associated with low circulating pro-NT levels and predicts lower insulin resistance in obese and overweight children [129] These observations confirm that decreased *MMP7* expression is deleterious and results in aberrant gut microbiota composition, enhanced intestinal fat absorption and ultimately leads to obesity and associated metabolic abnormalities such as NAFLD. Additional work is necessary to dissect the exact role of MMP7 in adipose tissue homeostasis. Is the NT/MMP7/DEFA5 axis deregulated in adipose tissue during human obesity development? This possibility is worth exploring, given that NT receptor 3 (NTS3)/sortilin is expressed in adipocytes and located in the same intracellular vesicles expressing the glucose transporter 4 (GLUT4). Importantly, NTS3 is translocated to the cell surface in response to insulin signaling [130,131]. The impact of obesity on the NT/NTS3 signaling pathway needs to be investigated in the future.

#### 2.5.2. MMP26

MMP26, also known as matrilysin-2, identified in 2000 as a protease lacking the hemopexin domain (like MMP7) endowed with a proteolytic activity on gelatin and β-casein [132].

Study of MMP26 mRNA steady states levels revealed a specific expression in placenta and epithelial glandular cells in the normal endometrium [132,133].

In disease, it has been mainly studied in cancer and tumor invasiveness, and has been recently described with MMP7 as additional biomarkers (with the well-established CA 15.3) for breast cancer diagnosis and progression [134], but also as a potential biomarker for prostate cancer [135]. In metabolism, a variant for *MMP26* (rs2499953) is associated with higher fasting glucose levels and identified as a deleterious risk allele for diabetes in a general French population (D.E.S.I.R. study), although only a modest association was found with a higher incidence of T2D [136]. In a previous French case-control study, the rs2499953 allele was also identified with a higher frequency in non-obese (BMI <30 kg/m^2^) diagnosed type 2 diabetic patients who had at least one 1^st^ degree relative with T2D compared with control non-diabetic individuals [137].

The contribution of the *MMP26* variant to T2D incidence should be confirmed in a larger cohort of patients and perhaps in another population of patients worldwide; and if so, investigate the potential role of MMP26 in T2D pathogenesis.

### 2.6. MMP19

MMP19 cleaves the laminin five gamma two chain and induces epithelial cell migration. *MMP19* is widely expressed in human tissue with a strong expression in the intestine, ovary, spleen, pancreas, lung and placenta, suggesting its important role in these tissues [138].

*MMP19* deficient mice are viable, have a normal development and are fertile, yet aged mice exhibit a higher body weight compared to controls on a standard diet [139]. However, when challenged with a HFD, *MMP19*^−/−^ animals gain significantly more weight in comparison to WT and have markedly increased weight of both ScWAT and GWAT. Moreover, these mice were less susceptible to skin cancer development [139]. Paradoxically, *MMP19* was shown to be markedly induced in later stages of adipocyte differentiation and in adipose tissue of mouse models of obesity [31]. One explanation could be that MMP19 may negatively control adipogenesis and the increased expression in obesity may be a way of limiting further weight gain. Interestingly, *MMP19* is expressed in both adipocytes and cells of the SVF [31]. Moreover, *MMP19* was identified, among other MMPs, by microarray analysis to be robustly upregulated in human adipocytes treated for 4 h or 24 h with macrophage-conditioned medium [69], suggesting a paracrine regulatory role of MMP19 during adipose tissue expansion in obesity. The molecular mechanisms at play remain poorly defined, and other studies are required to determine precisely the role of MMP19 in adipocyte-SVF cells crosstalk during obesity.

### 2.7. Membrane-Type MMPs

MT-MMPs are membrane-anchored proteases by a transmembrane domain (MT1, MT2, MT3, and MT5-MMP) or a glycosyl-phosphatidyl-inositol (GPI) anchor (MT4- and MT6-MMP) and have various functions related to ECM degradation, protein cleavage and release of signaling molecules [140]. The membrane localization of MT-MMPs leads to the concentration of receptors in specific regions, resulting in increased pericellular proteolysis. MT-MMPs have a wide range of known substrates including fibronectin, vitronectin, laminin, tenascin, and proteoglycans-collagen. MT-MMPS with a transmembrane domain interact as well with scaffold proteins or cytoskeleton through their C-terminal tail. They are known to activate pro-MMP2 whereas GPI-anchored MT-MMPs do not. Thus, they play crucial regulatory roles both in cell-cell communication and intracellular signaling [140]. Pathologically, MT-MMPs are important contributors to cancer progression (see review in Ref. [141]). Their roles in metabolic functions and diseases have been recently unveiled, especially those of MMP14.

#### 2.7.1. MMP14 (MT1-MMP)

MMP14 plays a crucial role in ECM homeostasis and remodeling through two distinct mechanisms. Firstly, it facilitates the activation of proMMP2 by interacting with TIMP2, which acts as a bridge between proMMP2 and MMP14. This interaction allows for the activation of pro-MMP2 [142]. Secondly, MMP14 exhibits inherent proteolytic activity towards ECM components independent of MMP2 activation [143,144]. These dual actions highlight the pivotal role of MMP14 in maintaining the balance and integrity of the ECM. Recently, the involvement of MMP14 in ECM remodeling during obesity has been unveiled.

##### MMP14 Allows Adipose Tissue Expansion during Development

Generation of genetic-engineered mice targeting *MMP14* brought novel insight into the role of MMP14 in adipose tissue development and homeostasis. Mice lacking *MMP14* exhibit a dramatic reduction in body size and premature death [145]. Glycogen and lipid levels are significantly lower in *MMP14*^−/−^ mice due to increased levels of rate-limiting enzymes involved in catabolic pathways compared to wild-type (WT) animals. Hence, *MMP14*^−/−^ neonate animals have significantly reduced circulating triglyceride and glucose levels compared with controls. Moreover, *MMP14* silencing in mammary gland epithelial cells induces enhanced autophagy. These observations explain the failure of *MMP14*^−/−^ neonates to thrive and their premature death due to a profound metabolic disorder [145]. Another study reported the importance of MMP14 in collagen I turnover. Upon a HFD, *MMP14* haplo-insufficient mice are unable to gain weight because of their failure to remodel fat pad collagen architecture [146]. These two studies underline the primordial role of MMP14 in adipose tissue remodeling and metabolism.

##### MMP14 Is Upregulated during HFD

Li et al. reported a robust increase in *MMP14* and collagen VI expression (both mRNA and protein levels) in obese adipose tissue of mice as early as 5 weeks of a HFD and this expression was more pronounced after 18 weeks of high fat feeding compared to lean adipose tissue [147]. Interestingly, MMP14 was shown to be induced by hypoxia and treatment with a Hypoxia Induced Factor 1α (HIF1α) inhibitor prevents *MMP14* induction in vitro, while chromatin immunoprecipitation experiments demonstrated that HIF1α binds with a high affinity to three consensus motifs in the *MMP14* promoter [147]. Moreover, adipose tissue-specific induced overexpression of *MMP14* in established obese adipose tissue resulted in increased body weight, adipocyte hypertrophy, fatty liver, insulin resistance and impaired energy expenditure, confirming the prominent role of MMP14 in obesity and associated abnormalities [147]. Furthermore, *MMP14* overexpression promotes adipose tissue fibrosis and inflammation with a switch in M2 to M1 macrophage polarity. Mechanistically, MMP14 digests collagen type VI alpha 3 chain, to produce endotrophin, a potent inducer of fibrosis and inflammation [147].

Consistent with the aforementioned observations, *MMP14* has been shown to be induced in adipose tissue of genetic mouse models of obesity (*Ob/Ob* and *db/db* mice) and this expression correlates with the presence of active MMP2 [31]. In fact, MMP14 was previously shown to activate proteolytically pro-MMP2 [143,144]. In addition to MMP2, our laboratory has demonstrated that MMP14 cleaves the catalytic domain of MMP11 [142], an important regulator of energy metabolism protecting from diet-induced obesity, fatty liver and insulin resistance [76].

In humans, genetic variants located near the catalytic domain of *MMP14* were associated with obesity traits due to enhanced collagen turnover [146].

Altogether, these studies highlight the dual action of MMP14 during nutritional stress: at first, MMP14 is induced via the hypoxic pathway to digest collagens and limit fibrosis, meanwhile activating MMP2 and hindering MMP11, thereby allowing the expansion of fat tissue. At later stages of obesity, a high quantity of endotrophin is produced by the MMP14-induced digestion of COL6α3, which has pro-inflammatory and pro-fibrotic actions, contributing to insulin resistance.

#### 2.7.2. MMP15 (MT2-MMP)

The MMP15 gene is located in the chromosomal region 16q12.1, adjacent to the heterochromatin [148]. While MMP15 has the ability to cleave collagen I, its specific activity is considerably lower compared to MMP14 [140]. MMP15 is known for its capacity to cleave the NC1 domain of collagen IV, releasing biologically active NC1 fragments. In addition, similar to MMP14, MMP15 can activate proMMP-2, which may explain its role in degrading the basement membrane and its implications in carcinogenesis [149,150].

MMP15 has also been implicated in human obesity and insulin resistance. Indeed, *MMP15* is significantly downregulated in both omental (OM) (~50% decrease) and ScWAT in obese patients with high- and low-insulin resistance status (~40% and ~33% decrease, respectively) compared to lean individuals. Consistently, a clear negative correlation is observed between *MMP15* expression and HOMA-IR index in both tissues [41]. Moreover, *MMP15* was shown to be dramatically downregulated in TNFα-treated human SGBS human adipocytes compared to untreated adipocytes [151]. Given the contributing role of TNFα in insulin resistance development during obesity (see book chapter in ref. [51]), downregulation of MMP15 may act downstream of TNFα in obesity-associated low grade inflammation to promote insulin resistance. More work is required to fully unravel the exact role of MMP15 depletion in obesity and insulin resistance states.

Finally, *MMP15* transcripts (with calpain-4 and calpastatin) were shown to be significantly increased in subcutaneous adipose-derived stromal/stem cells (ASCs) from obese individuals and demonstrated increased invasion through Matrigel and chick chorioallantoic membrane, a type I collagen-rich extracellular matrix barrier, compared to non-obese subjects. Knockdown of *MMP15* yielded decreased invasion, demonstrating the implication of MMP15 and obesity status in the invasive capacity of ASCs [152], which have been associated with increased breast cancer tumorigenesis and metastasis [153,154]. This study identifies MMP15 as a protease that may allow ASCs egress through ECM barriers and invasion of surrounding tissue in the context of breast cancer. Other studies are still required to understand the molecular mechanisms behind MMP15-mediated invasion of ASCs in breast cancer.

### 2.8. Tissue Inhibitors of MMP (TIMPs)

The TIMP family consists of TIMP1 to TIMP4, which are proteins comprising 184–194 amino acids and weighing 21–28 kDa. TIMPs are predominantly found in the extracellular matrix (ECM) in a soluble form, except for TIMP3, which binds to the ECM [4].

TIMPs’ binding to MMPs is specific and reversible, forming a 1:1 stoichiometric complex. TIMPs possess an inhibitory domain, located in the N-terminal domain, which binds to the catalytic domain of MMPs. The C-terminal tail interacts with the MMP hemopexin domain, providing additional stability to the TIMP-MMP complex. Although TIMPs have a high level of sequence similarity, there are differences in specificity for MMPs [155]. One TIMP targets multiple MMPs, as well as ADAM and ADAMTS, thus controlling a pyramidal cascade with complex context- and tissue-specific biological outcomes [19].

Furthermore, the function of TIMPs extends beyond MMP inhibition. Beyond structural changes in the ECM, TIMPs also determine the influence of the ECM on cell phenotype and proliferation, via their binding to cell adhesion molecules, cytokines, growth factors, and thereby they have a crucial role in ECM homeostasis, as well as cancer initiation and progression.

#### 2.8.1. TIMP1

TIMP1 (as TIMP3) has a glycosylated C-terminal tail. TIMP1 is a poor inhibitor of membrane-type metalloproteinases, such as MMP14 (MT1-MMP), but is a strong inhibitor of the majority of other MMPs, such as MMP1, 2, 3, 7 and 9. TIMP1 also interacts with proMMP9 in a non-inhibitory manner, through only their C-terminal domains [19].

##### TIMP1 Association with Metabolic Syndrome

First, it was already mentioned earlier that circulating concentrations of TIMP1 (and TIMP2) are significantly increased in patients with metabolic syndrome (MeS), and these levels are even higher in the presence of diabetes [44]. Moreover, TIMP1 and TIMP2 are correlated with BMI, waist circumference and metabolic parameters in all MeS patients [44]. The elevation in TIMP1 (and TIMP2) concentrations follows that of MMP2 and MMP9, perhaps as a way to limit the activity of these MMPs that could be deleterious.

##### TIMP1 Role in Adipocyte Differentiation

Wolfrum and coworkers reported an increase in both TIMP1 serum concentrations and adipose tissue protein levels in mouse models of obesity compared with lean control animals [156], confirming earlier results showing increased *TIMP1* expression in obese adipose tissue of Ob/Ob, db/db and mice submitted to a HFD [31]. Meissburger et al. demonstrated also that HFD-fed mice treated with murine recombinant TIMP1 (mrTIMP1) display enlarged adipocytes and a disturbed metabolic profile including enhanced circulating FFA and adipose tissue insulin resistance, yet bodyweights were not different between mrTIMP1-injected mice and controls [156].

Consistent with the role of TIMP1 as a positive regulator of adipogenesis, Lijnen et al. reported significantly lower gain weight, decreased ScWAT and GWAT fat pad weights and reduced adipocyte size in *TIMP1*^−/−^ mice, but also improved circulating metabolic parameters, when subjected to a HFD, whereas subtle and non-significant differences were observed on a standard diet [157]. No change in food intake nor in energy expenditure were reported under a HFD and a standard diet, and the changes observed on a HFD were ascribed to a decreased feeding efficiency (weight gain for food consumed) in *TIMP1*^−/−^ compared to *TIMP1*^+/+^ mice [157]. In agreement with these results, another study showed that *TIMP1*^−/−^ mice gained less weight compared to control mice when fed a high fat diet, but not intermediate fat-sucrose diet (IFSD) and were protected from HFD- and IFSD-induced glucose intolerance, hepatic steatosis and altered gene expression involved in lipid metabolism and inflammation. However, *TIMP1* deficiency did not alter insulin sensitivity nor insulin secretion under these conditions. Overall, the authors suggested TIMP1 is a contributing factor to diet-induced obesity, hepatic steatosis, and impaired glucose tolerance [158].

Interestingly, the reduced weight gain was not due to enhanced energy expenditure because of unaltered oxygen consumption, heat production and unchanged *Ucp1* and *Ppargc1a* (encoding the metabolic regulator PGC1α) adipose tissue expression [156,158], but rather due to decreased fat absorption as evidenced by enhanced energy content in the feces of *TIMP1*^−/−^ mice under a HFD [158]. Consistently, TIMP1 injection neither altered oxygen consumption, physical activity, and feeding [156]. Hence, TIMP1 may promote dietary fat absorption in conditions of nutrient excess without affecting energy expenditure. Most importantly, no changes in body weight and fat pads were observed in *TIMP1*^−/−^ mice fed a chow diet [157,158], strongly in favor of a role of TIMP1 in promoting fat uptake in conditions of nutrient excess. It remains to be determined, whether this effect is mediated by MMP-dependent or MMP-independent mechanisms.

Alexander et al. reported in vivo that *TIMP1* overexpressing (TO) mice display accelerated adipocyte differentiation and uncommon colonization of adipocytes in post-lactation involuting mammary gland. Thus, a 30–40% increase in adipocyte number and size was observed in involuting mammary gland of TO mice during active recolonization [59]. Treatment of committed 3T3-L1 pre-adipocytes with human recombinant TIMP1, with purified human TIMP1 or with a synthetic MMP inhibitor (Ilomastat, GM6001) accelerated the rate of differentiation compared with cells treated with dexamethasone alone [59]. In apparent contradiction, Meissburger et al., showed that mouse recombinant TIMP1 dose-dependently inhibits cell differentiation in 3T3-L1 and subcutaneous (but not visceral) primary pre-adipocytes, while neutralizing TIMP1 with an antibody results in enhanced adipocyte differentiation [156]. This is consistent with the high level of expression of *TIMP1* (and *TIMP3*) in committed pre-adipocytes and the low expression level observed in differentiated adipocytes [59], suggesting that TIMP1 may be important for pre-adipocyte proliferation and commitment, but its expression must be shut down later on to allow proper adipocyte differentiation.

##### TIMP1 and Hypothalamic Leptin Signaling

The role of TIMP1 was further complexified by the apparently contradictory results of the knock-out (KO) study by Gerin et al. who reported that, on a chow diet, deletion of *TIMP1* resulted in increased body weight at 3 months (Mo) of age in female mice compared to WT animals and this increase was maintained until 12 Mo due to enhanced body fat mass [159]. *TIMP1*^−/−^ mice displayed increased WAT and BAT weights, with an increase in the size and number of adipocytes but also an 11% increase in the bone mineral density compared to WT animals. Moreover, *TIMP1*^−/−^ mice exhibited hyperphagia despite hyperleptinemia suggesting these mice were leptin resistant and might harbor impaired hypothalamic leptin signaling [159].

Indeed, this phenotype and the increased fat mass may be ascribed to disturbed food intake behavior that is mediated by abnormal leptin signaling when *TIMP1* is inactivated. In fact, leptin positively regulates *TIMP1* expression in the hypothalamus via signal transducer and activator of transcription-3 (STAT3) signaling [159]. Mice lacking leptin inhibitory signals (l/l mice, mutated leptin receptor Y985L) [160] have a significant increase in hypothalamic *TIMP1* expression levels [159]. Therefore, TIMP1 appears to mediate the effect of leptin on food intake inhibition. Hyperphagia observed in *TIMP1*^−/−^ animals may be secondary to the described decrease in hypothalamic mRNA levels of *Hcrt* (encoding hypocretin) [159], a neuropeptide expressed in lateral hypothalamic area (LHA) that promotes hyperphagia acutely. Consistently, decreased *Hcrt* expression has been associated with weight gain in rodent models of obesity, in agreement with the documented long term catabolic action of HCRT [161,162,163]. If these results are confirmed, TIMP1 may play an important role in the rapid rewiring of neural circuits (orexigenic and anorexigenic neurons) in the hypothalamus induced by leptin treatment in *Ob/Ob* mice [164]. Of particular note, besides the role of TIMP1 in the regulation of food behaviour in the hypothalamus, *TIMP1*^−/−^ animals have impaired learning and memory, underscoring the important role of TIMP1 in the regulation of neuronal function and plasticity [165,166], but also other important functions such as in reproduction and heart function (for review see [155]).

It is worth mentioning that TIMP1 inactivation-induced hyperphagia and its subsequent effect on weight gain may mask the real impact of TIMP1 inhibition on adipose tissue differentiation/adipocyte size under conditions of nutritional excess. Gerin et al. also suggested that the effect of TIMP1 on the regulation of food intake may be gender dependent (no difference in body weight was reported in males), an assumption that was not verified by other studies. Differences in mice genetic background may account for the discrepancy between the studies of Gerin et al. (129S4/SvJae) and those of Fjære et al. (BALB/c) and Lijnen et al. (C57Bl/6 × 129 SvJae founders). Furthermore, Gerin et al. did not investigate the impact of a HFD in *TIMP1*^−/−^ mice compared to WT animals. Two questions arise: (i) what is the impact of a HFD on *TIMP1*^−/−^ mice in this genetic background (129S4/ SvJae)? and (ii) would hyperphagia be exacerbated in *TIMP1*^−/−^ mice upon a HFD? Much work is needed to understand the complex role of TIMP1 in the regulation of feeding and energy homeostasis in different organs. For this purpose, adipose tissue-, hypothalamic-, liver- and muscle-specific inactivation/overexpression of *TIMP1* in animals may bring new insight regarding the precise metabolic role of TIMP1 in each of the cited tissues and its impact on whole body energy homeostasis.

#### 2.8.2. TIMP2

TIMP2 is a ubiquitous, constitutively expressed, non-glycosylated protein. It is a highly effective inhibitor of multiple MMPs, including MMP2 and MMP9. Notably, TIMP2 forms a complex with MMP14 (MT1-MMP) and proMMP2, facilitating the activation of proMMP2 into its active form. This ternary complex is crucial for the controlled proteolysis of extracellular matrix components. These mechanisms highlight the pivotal role of TIMP2 as a key regulator of ECM remodeling and maintaining tissue homeostasis [19].

##### TIMP2 Association with Obesity and Metabolic Syndrome

As mentioned earlier, like TIMP1, TIMP2 levels were also shown to be increased in the serum of patients with MeS and were correlated with BMI [44]. In addition to *TIMP2* deficiency-induced neuromotor deficit [167] and defective myogenesis [168,169], *TIMP2*^−/−^, both male and female, mice exhibit obesity with a relatively preserved glucose tolerance and insulin sensitivity. However, upon a HFD, although obesity was exacerbated and glucose tolerance decreased in *TIMP2*^−/−^ mice from both sexes (male were more intolerant to glucose than female mice), only male *TIMP2*^−/−^ mice were insulin resistant with increased pancreatic β-cell mass and hyperplasia compared to controls [170], suggesting sex-diet interactions when TIMP2 is inactivated. Interestingly, expression levels of the glucose transporter *Glut2* and the pancreatic and duodenal homeobox 1 (*Pdx1*) transcription factor were decreased in *TIMP2*^−/−^ mice both under a chow and HFD compared with control mice [170]. Despite enhanced β-cell mass, male *TIMP2*^−/−^ mice under a HFD develop T2D, probably because of exhaustion of the β-cell, as demonstrated by the decrease in insulin immunostaining in pancreatic islets in comparison to WT mice. These results point to an important role of TIMP2 in insulin secretion and glucose homeostasis.

*TIMP2* expression is decreased in both ScWAT and visceral fat upon a HFD in males only (*TIMP2* expression was higher in ScWAT compared to visceral fat in males, whereas comparable expression is observed in female mice). The reason behind the enhanced HFD-induced insulin resistance development in *TIMP2*^−/−^ male mice may lie in the increased adipose tissue inflammation and macrophage infiltration compared to controls [170]. Moreover, collagenolytic activity is increased in ScWAT of *TIMP2*^−/−^ male mice as shown by increased active MMP14 (MT1-MMP) expression, in favor of an inhibitory effect of TIMP2 on MMP14. The phenotype observed in *TIMP2*^−/−^ mice may be MMP14-dependent as it has already been shown that upon a HFD, weight gain is blunted in mice with *MMP14*-haploinsufficiency compared to mice with normal *MMP14* expression due to a defect in fat pad collagen turnover as mentioned earlier (see Section 2.7.1) [146].

##### TIMP2 and Hypothalamic Leptin Signaling

*TIMP2*^−/−^ mice were found hyperphagic before the onset of obesity and were leptin resistant as evidenced by reduced leptin signaling in the arcuate nucleus of 2 month-old *TIMP2*^−/−^ mice fed a chow diet compared to control mice as well as decreased response to leptin administration on food consumption, as evidenced by decreased leptin-mediated STAT3 activation [171]. Moreover, increased net proteolysis was noted in the hypothalamus, but not globally in the entire brain, of *TIMP2*^−/−^ mice compared to WT animals. Notably, a significant reduction in the expression of the leptin receptor *ObRb* in the arcuate nucleus of *TIMP2*^−/−^ mice was observed at the start of the study before the installation of obesity, which strengthens the notion that reduced leptin signaling may reflect hyperphagia of *TIMP2* deficient mice. As mentioned earlier (see Section 2.1.1), reduced *ObRb* expression may be a result of increased MMP2-mediated ObRb cleavage [29]. Counterintuitively, expression of anorexigenic peptides was unchanged, while expression of the orexigenic neuropeptides NPY and AgRP was even decreased in the neonatal period [171]. It is possible that hyperphagia in *TIMP2*^−/−^ may be due to lesions in other areas of the hypothalamus. Indeed, hyperphagia and obesity have been previously shown to occur even when hypothalamic *Npy* expression levels are low in models with disrupted ventromedial or paraventricular nuclei as well as when communication with hindbrain autonomic centers is altered [172]. Obesity may also result from decreased leptin-mediated energy expenditure, and increased food consumption may be a counterregulatory mechanism secondary to heat loss. Finally, epigenetic modifications without altered gene expression were also advanced as a possibility not to be excluded to explain this phenotype [173]. Other studies are still required to help explain the precise role of TIMP2 in the regulation of food intake and energy homeostasis.

#### 2.8.3. TIMP3

TIMP3 is glycosylated at the C-terminal tail. Unlike other TIMPs, which are soluble, it is tightly bound to the ECM. Among TIMPs, TIMP3 exhibits the broader inhibitory specificity towards multiple members of the MMP family, including MMP1, MMP2, MMP3, MMP9, and MMP14. Additionally, TIMP3 is capable of inhibiting various ADAMTS enzymes (A Disintegrin and Metalloproteinase with Thrombospondin Motifs). The broad inhibitory profile of TIMP3 highlights its crucial role in regulating locally MMP-mediated proteolytic processes, thereby maintaining the balance of extracellular matrix remodeling in diverse biological contexts [19].

TIMP3 has a protective role in obesity-related metabolic abnormalities. Differential display experiments on mRNA extracted from the muscle of insulin receptor haplo-insufficient mice (*Insr*^+/−^) with or without insulin resistance mice identified a marked decrease in *TIMP3* expression in *Insr*^+/−^ mice with insulin resistance compared to insulin sensitive animals. The lower *TIMP3* expression was responsible for the increase in tumor necrosis factor α converting enzyme (TACE) activity and subsequent elevation in soluble TNFα in serum and muscle inducing vascular inflammation, hyperglycemia and hyperinsulinemia as compared with *Insr*^+/−^ mice that were normoglycemic and had unchanged *TIMP3*. Consistently, double heterozygote *Insr*^+/−^/*TIMP3*^+/−^ mice were overtly hyperinsulinemic and hyperglycemic and had robust increased serum and muscle TNFα levels at 6 months of age [174]. This phenotype is also partly due to enhanced adipose tissue hypertrophy, inflammation, and hepatic steatosis in *Insr*^+/−^/*TIMP3*^+/−^ mice compared to WT and single heterozygote *Insr*^+/−^ or *TIMP3*^+/−^ mice [175]. Interestingly, anti-TNFα therapy drastically improved glucose homeostasis in *Insr*^+/−^/*TIMP3*^+/−^ mice through decreased TNFα and MMP levels [174]. Hence, inhibition of TACE/MMP activities may represent a viable therapeutic approach to counter insulin resistance and diabetes.

Conversely, macrophage-specific *TIMP3*-overexpressing (*MacT3*) mice were protected from HFD-induced metaflammation, insulin resistance, glucose intolerance, and NASH through decreased oxidative stress-related pathways, yet their body weight did not differ much from control mice upon a HFD consumption [176]. These mice displayed reduced size of atherosclerotic lesions with enhanced stability, lower inflammation, and improved metabolic parameters when crossed with low-density lipoprotein receptor (LDLR)-deficient (*Ldlr*^−/−^) mice [177]. Increased delivery of TIMP3 may be tested in preclinical models of diabetes and atherosclerosis, especially because *MacT3*/*Ldlr*^−/−^ mice had a lower body weight than *Ldlr*^−/−^ mice under a HFD [177].

#### 2.8.4. TIMP4

Unlike other members of the TIMP family, TIMP4 is typically absent or present at very low levels in most tissues. TIMP4 shows the highest homology with TIMP2. Similar to TIMP2, TIMP4 can form a ternary complex with MMP14 (MT1-MMP) and proMMP2. However, unlike TIMP2, the binding of TIMP4 to this complex does not result in the activation of proMMP2. Instead, TIMP4 competes with TIMP2 for binding to the complex, potentially exerting an inhibitory effect on MMP2 activation [19,155].

##### TIMP4 Expression during In Vitro Adipocyte Differentiation

The role of TIMP4 is complex, controversial, and not fully understood as it is tightly linked to MMP3. *TIMP4* was shown to be highly expressed in the adipose tissue and its expression level is almost undetectable in 3T3-L1 pre-adipocytes, weak in committed pre-adipocytes and rises further upon induction of differentiation in parallel with expression of the master differentiation factor PPARγ [59,178]. Notably, Wu et al. reported that in vitro MMP3 overexpression in 3T3-L1 cells or cells treated with human recombinant MMP3 significantly reduced adipogenesis, whereas concomitant treatment of these cells with recombinant TIMP4 was shown to improve adipogenesis and decrease the inhibitory effect of MMP3 [58]. These observations suggest that in vitro, MMP3 has an inhibitory role on adipogenesis, whilst TIMP4 has a stimulatory effect.

Unexpectedly, Mejia-Cristobal et al. showed also that *TIMP4* silencing using short hairpin RNA (shRNA-*TIMP4*) accelerates late phase adipocyte differentiation through decreased NFκB activity contributing to the terminal differentiation burst [178]. Moreover, microarray analyses of shRNA-*TIMP4* 3T3-L1 adipocytes showed that the NFκB pathway is among the top regulated pathways [178]. This study revealed that upon TIMP4 inactivation, the expression levels of *Pparγ* and *Cepbα* are significantly enhanced [178]. Activation of the NFκB pathway has previously been shown to inhibit adipogenesis through decreased expression of *Pparγ* [179]. Altogether, it is plausible that TIMP4 may have a dual role in adipogenesis. TIMP4 may be pro-adipogenic during the early phase of differentiation to block MMP3, and its increased expression during terminal differentiation may serve as a negative regulator to avoid excessive fat accumulation. Nevertheless, given the complexity of adipose tissue and the interconnexion of adipocytes with cells of the stromal vascular fraction, these regulatory roles of TIMP4 and MMP3 need to be studied in vivo and in different fat pads in standard conditions and upon a nutritional challenge.

##### In Vivo Expression of TIMP4 Is Gender- and Fat Depot-Dependent

The regulatory roles of TIMP4 and MMP3 have been studied in vivo and in different fat depots sites, in standard conditions and upon a nutritional challenge by Wu et al. [58]:Differential expression of TIMP4 depending on fat depot site

Under standard diet (SD) conditions, TIMP4 expression did not differ between GWAT and inguinal ScWAT in females. In males, TIMP4 transcripts tend to be lower in inguinal ScWAT compared to GWAT.
2.Differential expression of TIMP4 depending on sex

Under SD conditions, TIMP4 expression levels were lower in all depots (ScWAT and visceral fat) in females compared to males, but this difference was lost after a high-fat diet (HFD) feeding [58].
3.Influence of HFD

Under a HFD, TIMP4 expression was increased in all fat depots in females whereas in males TIMP4 transcripts were significantly decreased in all fat pads, underscoring significant sex/diet interactions in all fat depots [58].

As mentioned earlier, the gender- and fat depot-dependent regulatory roles of TIMP4 in adipogenesis need to be linked to those of MMP3 and one may rather reflect in terms of a MMP3/TIMP4 ratio. For instance, the fact that *MMP3* is mainly expressed by mouse APCs and human pre-adipocytes and that *TIMP4* is mainly expressed by adipocytes, may represent a key element in the differences observed between males and females. Indeed, recruitment of APCs is important in adipose tissue expansion through hyperplasia and male gonadal fat has a limited capacity for hyperplasia and expands mainly by hypertrophy [180,181,182,183]. It appears that downregulation of *MMP3* is critical to allowing APCs to fully differentiate and accumulate lipids in the presence of the proper adipogenic signal. Upon a HFD, downregulation of APC *MMP3* is important to trigger adipogenesis, whilst adipocyte TIMP4 may be important for the regulation of hyperplasic vs. hypertrophic tissue expansion and sex-dependent fat distribution. Hence, the balance between adipocyte *TIMP4* expression and APC production of MMP3 determines sex- and depot-dependent growth of adipose tissue (hyperplasia vs. hypertrophy) in response to developmental and environmental cues.

##### TIMP4^−/−^ and TIMP4 Overexpressing Mouse Model Phenotypes

The role of TIMP4 in adipogenesis needs to be understood in vivo, where different cell types interact to achieve adipose tissue homeostasis in the whole body. The *TIMP4*-loss-of-function mouse model ascribed an obesogenic role for TIMP4 and a regulator of substrate utilization and energy homeostasis upon nutrient insult [184]. Indeed, *TIMP4* deficiency did not affect body weight under a chow diet compared to WT animals, whereas under a HFD, *TIMP4*^−/−^ mice had a significantly lower body weight, weight gain, and fat mass (mainly epididymal) than their control counterparts despite increased food intake, while the percentage of body lean mass was increased under a chow diet and HFD *TIMP4*^−/−^ mice compared to WT counterparts both under chow and HFD [184]. More specifically, under a chow diet, despite a decrease in body fat percentage, the size of white adipocytes in epididymal fat was increased in *TIMP4*^−/−^ compared to WT, that could reflect an impaired pre-adipocyte recruitment/proliferation and accumulation of lipids in existing mature adipocytes in this tissue. However, under a HFD TIMP4^−/−^ mice displayed smaller adipocytes, reduced WAT inflammation and collagen deposition [184]. Notably, *TIMP4*^−/−^ mice were protected from a diet-induced increase in circulating FFA and LDL-cholesterol and displayed decreased liver and muscle triglyceride accumulation. This effect was ascribed to defective lipid absorption and increased FFA excretion because of decreased intestinal fatty acid transporter CD36 protein levels, while brown adipose tissue (BAT) was unaltered and basal metabolic rate as well as energy expenditure were reduced in *TIMP4*^−/−^ mice compared to WT animals [184]. As mentioned earlier in the study by Wu et al. 2017, the ratio MMP3/TIMP4 is important in regulating fat depots. Hence, inactivation of TIMP4 is in favor of a high MMP3/TIMP4 ratio, consistent with weight loss.

This is in agreement with results from Alexander et al. showing that *MMP3*/stromelysin-1 deficiency yields accelerated mammary gland adipogenesis during involution, whilst *TIMP4* overexpressing mice display an enhanced number and size of adipocytes as early as 2 days after weaning [59], suggesting that an increased TIMP4/MMP3 ratio promotes increased fat mass through hyperplasia (increased number) and hypertrophy (increased size).

It Is worth mentioning that a HFD-*TIMPp4*^−/−^ mice were not protected from diet induced glucose intolerance, despite reduced insulinemia and decreased fat accumulation and inflammation. This may be due to an increased utilization of lipids and decreased utilization of glucose under a HFD. On the contrary, under a chow diet, a significant increase in respiratory exchange ratio (RER >0.9) was observed in *TIMP4*^−/−^ mice compared with WT mice [184], suggestive of increased carbohydrate utilization and decreased lipid oxidation [185]. Hence, in addition to its role in lipid absorption and effect on food intake, TIMP4 may be essential in the regulation of substrate utilization in response to a HFD that could partly explain the weight loss and the failure to improve glucose tolerance despite decreased insulin resistance.

More recently, a cross-sectional study including South-African individuals (546 patients with normal glucose tolerance, 116 patients with impaired glucose metabolism and 93 T2D patients) from the Middle-Aged Soweto Cohort (MASC) identified 73 proteins that were associated with dysglycemia, 34 of which were validated in the Swedish EpiHealth cohort. Among these, 11 proteins were associated with insulin dynamics measurement and TIMP4 was found to be significantly associated with higher insulin secretion and β-cell function in women only [186], highlighting again TIMP4-mediated sex-dependent differences related to glucose metabolism regulation in addition to those related to fat depots. Notably, a posteriori analysis showed that association of TIMP4 and insulin dynamics were not mediated by adiposity.

##### Future Perspectives for Studying TIMP4/MMP3 Role in Metabolism

Altogether, these observations converge to the notion that TIMP4 is an important factor for adipose tissue differentiation and lipid absorption in response to nutrient excess and governs, with MMP3, depot- and sex-dependent adipose tissue expansion during obesity. Finally, TIMP4 has a fat-independent role on insulin secretion and β-cell function in women, and its deregulation may have a profound effect on the incidence of T2D. More work is needed to understand the molecular events behind this phenomenon. Cell-specific modulation of TIMP4 and MMP3 in vivo and tissue-specific inactivation of these proteins using “floxed” mice will undoubtedly help understand the subtleties of the role of TIMP4 and MMP3 in lipid and glucose homeostasis both in physiological conditions and in response to nutrient surplus.

### 2.9. Conclusion on the Role of MMPs and TIMPs on Adipose Tissue Homeostasis and Related Diseases

MMPs and TIMPs play crucial roles in maintaining the balance and function of adipose tissue. They are actively involved in adipocyte differentiation, maturation, and the regulation of adipose tissue growth, including hyperplasia and hypertrophy (Figure 2). Impaired or increased expression and activity of MMPs and TIMPs are seen in related diseases, namely obesity, insulin resistance and diabetes. The complexity of studying MMPs and TIMPs stems from the diversity of MMPs and their natural inhibitors, their intricate interactions with each other, their multifaceted actions on adipose tissue at both local and distant levels, as well as the variations observed between sexes and different types of fat deposits.

## 3. MMPs and TIMPs in Atherosclerosis and Diabetes Vascular Complications

Besides their contribution in the pathophysiology of obesity, insulin resistance and diabetes, MMPs are also implicated in complications arising from these diseases. Atherosclerosis, diabetic nephropathy (DN), and retinopathy are among the most serious complications of both type 1 and type 2 diabetes [187,188].

There is a limited association between hyperglycemia and cardiovascular complications of diabetes, and increased risk of cardiovascular diseases in diabetic patients is not only explained by dyslipidemia and hypertension. Rather, insulin resistance and subsequent modification of extracellular matrix in atherosclerotic plaques, kidney capillaries, and myocardium seem to be major determinants of these complications [189].

### 3.1. Atherosclerosis

#### 3.1.1. MMP8

The role of MMP8 in the progression of atherosclerotic disease has been well studied. *MMP8* is overexpressed by macrophages, endothelial cells, and smooth muscle cells in progressing atherosclerotic lesions [190,191,192]. Plasmatic MMP8 concentrations were linked to the presence and severity of coronary artery disease [193] and serum MMP8 levels were associated with cardiovascular outcome in patients [194]. The recent discovery of small molecule inhibitors of MMP8 such as sulfated mimetics of glycosaminoglycans (sulfated quinazolinones) looks promising to prevent atherosclerosis-mediated cardiovascular events [195].

MMP8 was associated with lipid metabolism as it was related with MMP8-mediated proteolysis of apolipoprotein A-I, the major protein in HDL particles [196], thus reducing cholesterol efflux from cholesterol-loaded THP1 macrophages and leading to increased accumulation of cholesterol in the vessel walls. This effect was blocked by the antibiotic doxycycline, a broad-spectrum MMP inhibitor [197]. In vivo, *MMP8* deficient mice displayed lower TG levels and larger HDL particles but no difference in serum Apo A-I nor in cholesterol efflux capacities was noted compared to WT mice. This observation was explained by the fact that Apo A-I is less susceptible to degradation when associated with lipids as it is the case in lipid-enriched HDLs, while MMP8-mediated proteolysis is facilitated in a lipid-free environment or within Apo-AI-lipid discs with smaller amounts of phospholipids and cholesterol [196].

#### 3.1.2. MMP2 and MMP9

MMP2 and MMP9 concentrations and activities (zymography) were measured in the serum of T2D patients with or without peripheral artery disease (PAD) [198,199]. Plasma concentration of both MMP2 and MMP9, together with MMP9 activity, were increased in diabetic patients with or without PAD compared to the control. Notably, MMP2 activity was higher in diabetic patients with PAD compared to diabetic patients without PAD. These results indicate that deregulation of these MMPs and subsequent increase in proteolysis may contribute to the pathogenesis of diabetes vascular complications [200,201]. Importantly, MMP9 serum concentrations (but not MMP2) were previously shown to be increased in patients with cardiovascular disease due to acute myocardial infarction and MMP9 was proposed as a marker of inflammation in this disease [202].

#### 3.1.3. TIMPs

Adenovirus-mediated overexpression of *TIMP1* was shown to significantly reduce atherosclerotic lesions in *ApoE*^−/−^ mice under a HFD [203], whereas *TIMP2* overexpression, but not TIMP1, inhibits atherosclerotic plaque development and destabilization in *ApoE*^−/−^ mice under a HFD through modulation of macrophage and foam cell behavior [204].

The complex role of MMPs and their inhibitors in vascular intima thickening and atherosclerosis plaque rupture goes beyond the scope of this review, and we invite the reader to refer to a more comprehensive report on the subject [205].

### 3.2. Diabetic Nephropathy

Diabetic nephropathy (DN) is another major complication often associated with diabetes. Extracellular matrix accumulation and modification is a key morphologic feature of DN and several studies have shown a correlation between activity and expression of MMP and progression of diabetic nephropathy [206].

Rysz et al. measured the circulating levels of MMP2, MMP9 and their inhibitors (TIMP1 and TIMP2) in T2D patients with or without DN, in patients with non-diabetic chronic renal failure (CRF), and in healthy controls [207]. Diabetic nephropathy (DN) patients had a significant reduction in serum levels of MMP2, TIMP1 and TIMP2 compared to the controls. TIMP1 and TIMP2 were decreased in DN patients compared to either patients with T2D alone or CRF alone [207]. In addition, ~two-fold increase in MMP9/TIMP1 and MMP2/TIMP2 ratio were observed in DN patients when compared to T2D patients with normal function. Moreover, TIMP2 was decreased in T2D patients compared with CRF patients alone and MMP2 was decreased in both T2D and CRF patients compared to the controls. In summary, this study demonstrates that changes in circulating MMP/TIMP ratio in the course of diabetes is linked to ECM remodeling in the kidney and diabetic nephropathy development [207].

In a diabetic rat model, serum and urine concentrations of MMP9 increased in the very early stage of nephropathy, with a higher MMP9 expression level in the proximal tubular epithelial cells [208]. In another animal study, it was demonstrated that methylglyoxal-derived advanced glycated end-products (AGE-4), proteins that accumulate in the ECM during the progression of diabetes, induced a substantial increase in the expression of *MMP2* and *MMP9* in rat ordinary kidney cells through the AGE-4/RAGE axis activating the ERK/JNK/NFκB signaling pathway, which ultimately promotes kidney dysfunction [209]. These results strongly suggest that MMP2/9-induced ECM remodeling contributes to AGE-mediated kidney function impairment during diabetic nephropathy.

In a model of streptozotocin-induced T1D, DN was also associated with alterations in the expression of *MMP9* and *MMP2* and other key ECM mediators in the kidney [210].

In another study, the methylation status of a panel of genes that are closely related (*MMP2*, *MMP9*, *TIMP2*, *AKR1B1*, *MYL9*, *SCL2A4*, *SCL2A1* and *SCL4A3*), which were known to be involved in kidney development or diabetic kidney disease or those associated with dialysis-induced changes in gene expression in peripheral blood cells [211], has been studied in T2D patients with or without DN. Apart from the *MYL9* gene which was hypermethylated, the others were hypomethylated. A significant negative correlation between *TIMP2* and *AKR1B1* gene methylation and albuminuria levels was reported, suggesting that hypomethylation of these genes may represent early markers of DN [212].

Furthermore, it has been shown in rodent models that urinary enzymatic activity of excreted MMP2 and MMP9 increases in T1D mice compared to non-diabetic mice. This effect was shown to occur before hyperalbuminuria and could serve as an early biomarker of renal dysfunction in diabetes [213]. This observation was also translated in adolescents with early stage of T1D compared with non-diabetic controls [213]. These data suggest that MMP activities can be potentially translated to the clinic and may represent useful biomarkers to predict the occurrence of kidney injury and vascular complications before the onset of albuminuria in diabetic patients.

Consistent with these observations, *MMP9*-single nucleotide polymorphism (SNP) was recently shown to be independently associated with increased susceptibility to develop DN in T2D patients (odd ratio 6.07 [1.60–22.99], *p* = 0.008) [214]. Recently, SNPs affecting MMPs that belong to three different families of proteases have been associated with susceptibility to type 2 diabetes nephropathy in a population of 310 patients with type 2 diabetic nephropathy and 310 healthy controls. Functional variants in *MMP1* (-1607 1G/2G), *MMP2* (-1306 C/T) and *MMP3* (-1171 5A/6A) were demonstrated to affect the transcriptional activity of these MMPs and were associated with increased susceptibility to DN occurrence and progression of renal disease [215].

*MMP24* (*MT5-MMP*) expression was increased in the kidneys of diabetic patients and was shown to actively process pro-MMP2 [140,216,217]. *MMP24* expression was localized to epithelial cells from distal and proximal tubules, collecting duct and Henle’s loop; and tubular epithelial cells expressing *MMP24* were associated with tubular atrophy, a marked contributing factor for DN development and kidney failure [218]. MMP24-induced MMP2 activation may play an important role in kidney remodeling in diabetic patients. Additional work is needed to dissect the molecular events involved both upstream and downstream of MMP24 to induce diabetic nephropathy.

Finally, the GPI-anchored *MMP17* (*MT4-MMP*) was shown to be expressed in the glomeruli of streptozotocin-mediated diabetic nephropathy in rats compared to control rats [219].

### 3.3. Diabetic Retinopathy

Retinopathy is a common complication of diabetes. Notably, upregulation of thrombin/MMP1/protease-activated receptor-1 (PAR1) pathway in retinas of diabetic rats and in vitreous samples from patients with proliferative diabetic retinopathy compared to non-diabetic patients has been reported to promote angiogenesis and progression of proliferative diabetic retinopathy [220]. This pathway was also investigated in human retinal microvascular endothelial cells (HRMEC) exposed to high glucose concentrations. MMP2 and MMP9 have also been linked to diabetic retinopathy (DR) development. Increased *MMP2* and *MMP9* expression levels are found in the retina and vitreous in DR patients and animal models of DR. During the early phases of diabetic retinopathy, activation of both gelatinases occurs in the retina, which will damage the mitochondria and causes apoptosis of the retina capillary cells, whereas in the later stages they favor neovascularization, promoting thereby proliferative retinopathy [221,222]. A proposed mechanism relies on intracellular accumulation and activation of MMP2 and MMP9 [223].

### 3.4. Diabetic Cardiomyopathy

Diabetic cardiomyopathy, which is defined by the alteration of the ventricular function of diabetic patients in the absence of coronary atherosclerosis or hypertension, is associated with left ventricular hypertrophy, fibrosis, and change in stiffness.

In streptozotocin (STZ)-induced diabetic rats, cardiac fibrosis was associated with reduced MMP2 activity, increased *Smad7* and *TIMP1* and decreased *MMP14* (*Mt1mmp*) expression [224].

*MMP28*, which has been more recently identified, has an ubiquitous expression in adult tissues suggesting a role in tissue homeostasis [225]. Only two MMP28 substrates have been identified so far, namely casein and the Neural Cell Adhesion Molecule (NCAM) [225]. A recent work has demonstrated that deregulation of *MMP28* expression occurs in a rodent model of diabetes. Indeed, *MMP28* is markedly downregulated in the left ventricle of aged obese diabetic Zucker Diabetic Fatty (ZDF) rats compared to control lean rats, suggesting the role of MMP28 in heart function and a potential involvement of impaired MMP28 action in diabetic cardiomyopathy [226].

Whether the decrease in MMP expression is a cause or a consequence of altered left ventricular function remains to be determined.

## 4. Muscle Metabolism, Differentiation, and Regeneration

Another key metabolic organ, the skeletal muscle system, plays a critical role in energy production, glucose regulation, lipid metabolism, protein metabolism, resting metabolic rate, and heat production. Low-twitch myofibers have a high capacity for oxidation and primarily utilize fatty acids as a substrate for ATP production, while slow-twitch fibers display decreased oxidative capacity.

Myogenesis and muscle metabolic profile determination rely upon ECM remodeling, which is associated with diet-induced insulin resistance. For instance, insulin-resistant skeletal muscle has increased the deposition of collagens. In this section, we will discuss how MMPs and TIMPs influence muscle differentiation and regeneration.

### 4.1. MMP2 and MMP9

In a fish model (Piaractus mesopotamicus), comparison of red muscle (thin, superficial layer, aerobic metabolism with slow contraction) and white muscle (glycolytic metabolism with fast contraction), from juvenile and adult fishes, showed that higher MMP2 and MMP9 activity was associated with rapid muscle growth in juvenile age and were associated with a different red and white physiology [227].

Additionally, *MMP2* expression was shown to be significantly enhanced in the muscle 7 days following cardiotoxin-mediated injury in a murine model [228]. Hence, MMP2 may play an important role in muscle repair. Whether this effect is direct or through MMP14-mediated activation of pro-MMP2 needs further clarification [229]. Finally, it was previously demonstrated that during the degeneration-regeneration process after muscle injury in normal mice and in the Mdx mouse model, MMP9 expression correlates with the inflammatory response (early response) and is most likely activated by satellite cells, whereas MMP2 activation is connected to the regeneration of new myofibers staging from myogenic cell proliferation to migration and fusion [230].

### 4.2. MMP10

MMP10 was demonstrated to be necessary for efficient muscle regeneration in both models of muscle injury and in the muscle of dystrophin-deficient mdx mice. Hence, *MMP10* expression is induced in response to damage in the skeletal muscle following injury or disease in the dystrophic muscle. Indeed, MMP10 inactivation in the mdx muscle led to a deteriorated phenotype compared to the mdx muscle with wild-type (WT) *MMP10* expression. In addition, silencing of *MMP10* in injured muscles from WT and mdx mice significantly altered regeneration, whereas treatment with recombinant MMP10 accelerated the repair process [231].

### 4.3. MMP13

*MMP13* has been shown to be progressively induced during in vitro C2C12 myoblast cell differentiation and is expressed by proliferating myoblasts with a more pronounced expression during fusion of myoblasts and myotube formation [228]. Moreover, *MMP13*-stably transfected C2C12 cells display increased migration compared to control cells [228]. In vivo, *MMP13*^−/−^ mice exhibit lower body weight at 4 weeks of age, presumably due to skeletal developmental abnormalities at the growth plate and subsequent altered endochondral bone developments that were previously reported [232], yet under basal conditions, *MMP13*^−/−^ mice do not display skeletal muscle overt histological nor functional abnormalities. Hence, MMP13 is dispensable for muscle homeostasis [233]. However, under muscle injury following cardiotoxin injection, *MMP13* expression is transiently decreased (day 1 and d3 post injection), then induced at d7 and further increased at d11 post-injection [228], suggesting a functional role in muscle repair. Indeed, Smith et al. showed that 11 days after cardiotoxin injection, *MMP13*^−/−^ animals exhibit decreased tibialis anterior muscle fiber size area and have impaired muscle regeneration as demonstrated by the presence of necrotic fibers compared to WT animals [233]. In the mdx mouse model of chronic muscle dystrophy, mice exhibit increased expression of *MMP13* compared to C57/Bl6 mice [228], yet crossing *MMP13*^−/−^ mice with mdx does not result in an aggravation of the phenotype, a higher proportion of necrotic muscle fibers was reported though in the crossed mice compared to mdx mice. Moreover, *MMP13*^−/−^ animals show a significant decrease in muscle hypertrophic response to adenovirus-mediated IGF1 administration, which occurs through a satellite cell-independent mechanism [233]. In addition, myoblasts from *MMP13*^−/−^ mice display decreased migration capacity compared to WT mice. Finally, satellite cell-targeted inactivation of MMP13 results in postnatal growth impairment compared to WT mice and shows altered regeneration after injury recapitulating the phenotype of *MMP13*^−/−^ mice. These data show that MMP13 originating from progenitor cells is essential for normal repair following muscle injury [233].

### 4.4. MMP14

MMP14 (MT1-MMP) was identified as an essential contributor to myogenesis. During myoblast differentiation, *MMP14* is transiently and critically increased during the elongation process (48–72 h post differentiation induction) to gradually decline during the fusion phase. Alternatively, sh-*MMP14* expressing cells form fewer myotubes compared with control shRNA through defective fibronectin degradation [234]. Most importantly, *MMP14* deficient animals display altered heterogenous smaller myofibers with muscle tissue abnormalities at 4 weeks after birth, including accumulated abnormally processed laminin compared to control mice. In addition, mutant mice display some hypertrophied myofibers with centrally-placed nuclei reminiscent of muscle dystrophy. These data indicate that MMP14 could be regarded as a novel checkpoint myogenesis factor regulating the elongation phase for subsequent myotube formation probably to degrade ECM matrix that may prevent cell fusion, allowing thereby proper muscle differentiation and maintenance [234].

### 4.5. TIMP2

*TIMP2* expression is increased progressively during myoblast differentiation, and *TIMP2*^−/−^ mice exhibit altered myotube formation and decreased contractility and display signs of muscle weakness. TIMP2 is a regulator of myogenesis and neuromuscular junction development [168,169].

## 5. MMPs and TIMPs in Fatty Liver Disease

The liver’s metabolic functions are essential for regulating energy balance, maintaining blood glucose levels, and supporting overall metabolic homeostasis in the body.

Non-alcoholic fatty liver disease (NAFLD) develops when excessive fat accumulates in the liver (>5% of liver hepatocytes are affected) due to either defective fatty acid (FA) oxidation, increased FA synthesis, or both. Liver steatosis can be accompanied by inflammation and fibrosis, a condition named steatohepatitis (NASH). NAFLD has become endemic in the last years due to the increased incidence of obesity worldwide [235]. NAFLD can be caused by genetic factors, inappropriate dietary lifestyle (western nutritional habits) and epigenetic modifications [236]. Preclinical mouse models and clinical studies have significantly improved our understanding of the molecular events occurring during the development and progression of NAFLD [237]. In the last decade, multi-omic approaches helped identify new molecular targets in a search for a remedy against this disease that can ultimately lead to complications such as cirrhosis and hepatocellular carcinoma [238,239]. In this section, we discuss the role of MMPs and their inhibitors in the pathogenesis of NAFLD.

### 5.1. MMP2 and MMP9 and Their Inhibitors

Besides obesity and diabetes, serum concentration and expression levels of MMP2 and MMP9 have also been studied in non-alcoholic fatty liver disease (NAFLD). Indeed, serum concentrations of MMP2, MMP9, TIMP1, TIMP2, MMP7, TGF-β1 and -β2 were found to be significantly higher in NAFLD patients compared to controls, yet these markers are unable to distinguish between steatosis and steatohepatitis. Most interestingly, TIMP1 concentrations are higher in patients with significant fibrosis than those without. In fact, TIMP1 was found to be an independent predictor of fibrosis in NAFLD patients [240] and TIMP1 plasmatic levels were recently shown to be significantly correlated to the percentage of fibrosis in a mouse model of NASH [241]. Moreover, TIMP1 serum concentration serves along hyaluronic acid and procollagen type 3 N-terminal propeptide (PIIINP) to calculate the Enhanced Liver Fibrosis (ELF) score for advanced liver fibrosis assessment (ELF ≤ 9.8 to rule out advanced LF, ELF > 11.3 to rule in advanced LF) [242]. Although employed in many countries, the ELF score has not been yet approved by the food and drug administration (FDA) and needs further validation in patients with low/moderate prevalence of advanced fibrosis. Of note, other biological serum parameter-based fibrosis evaluation scores exist (e.g., FIB-4, FibroTest^®^, NAFLD fibrosis score) and some of them need to be interpreted with caution in younger patients and in T2D patients in whom they underperform [243,244].

Another group showed that TIMP1 and TIMP2 liver concentrations are significantly increased in patients with severe fibrosis as compared to those with mild or no fibrosis, while pro-MMP2 activity is significantly higher in the liver of patients with severe and mild fibrosis compared to patients with no fibrosis [245]. Moreover, pro-MMP9 was the only ECM component to be associated with the severity of inflammation in NASH patients and was not correlated to fibrosis [245]. This study highlights the association of α-smooth muscle actin (α-SMA), a marker of activation of hepatic stellate cells, pro-MMP2, TIMP1 and TIMP2 with severity of fibrosis in NAFLD patients, whereas pro-MMP9 is rather associated with the severity of inflammation [245]. Recently, our group reported a significant increase in *MMP9* transcripts, but not *MMP2*, in liver biopsies from obese NAFLD patients that underwent bariatric surgery as compared with samples from lean cholecystectomized patients. *MMP9* expression was significantly increased in both patients with steatosis and NASH [238]. Moreover, we observed a positive correlation between *MMP9* expression and *Il32* (interleukin-32), a proinflammatory cytokine, in liver biopsies from our cohort of patients [238]. Another study showed also increased expression of *MMP9*, and *MMP10* but not *MMP2*, in liver biopsies from patients with NASH, whereas the opposite was observed in patients with chronic viral-induced hepatitis (i.e., increased *MMP2* but not *MMP9*) [246]. However, another report showed higher *MMP2* liver expression, serum concentration of MMP2 and hyaluronic acid in patients with NASH compared to those with simple steatosis even when excluding patients with severe fibrosis [247]. In this study, it was highlighted that MMP2 may be a predictive factor of NASH even in patients with mild fibrosis since *MMP2* liver expression and MMP2 serum levels were still elevated in NASH patients without advanced fibrosis whereas serum hyaluronic acid concentration was not different in these patients as compared with patients with simple steatosis [247]. All these studies converge to the notion that MMP9 is rather associated with the severity of inflammation and MMP2 with fibrosis in patients, in agreement with data in preclinical models. Interestingly, a new mediator of hepatic fibrosis has been recently identified, insulin-like growth factor binding protein related protein 1 (IGFBPrP1) that regulates MMP/TIMP ratio. Knockdown of *Igfbprp1* expression in mice mitigates thioacetamide-induced hepatic fibrosis through Sonic Hedgehog (SHH)-mediated regulation of MMP2/TIMP2 and MMP9/TIMP1 balance, inhibition of hepatic stellate cell activation, decreased transforming growth factor beta (*Tgf1β*) expression, and degradation of the ECM [248].

### 5.2. MMP11

We have already mentioned earlier the prominent role of MMP11 in whole body metabolism through the regulation of the IGFBP1/IGF1/AKT/FOXO1 signaling pathway. Indeed, our group has demonstrated that mice deficient in *MMP11* develop features of the metabolic syndrome, with increased triglyceride accumulation in the liver reminiscent of hepatic steatosis. Consistently, we showed in a GOF mouse model that MMP11 protects against diet-induced obesity, insulin resistance and hepatic steatosis through increased fatty acid turnover and oxidation [76]. These observations clearly demonstrate that MMP11 inactivation contributes to the development of NAFLD. Apart from the IGFBP1/IGF1 axis, it remains to be determined whether MMP11 regulates other molecules involved in NAFLD pathogenesis.

### 5.3. TIMP3

It was mentioned earlier that TIMP3 decrease is responsible for enhanced muscle, vascular and adipose tissue inflammation in diabetic *Insr*^+/−^ mice compared to non-diabetic *Insr*^+/−^ mice and double heterozygote *Insr*^+/−^/*TIMP3*^+/−^ mice were insulin resistant and exhibit overt hyperglycemia [171]. In *Insr*^+/−^/*TIMP3*^+/−^ mice, the liver, a major organ involved in the pathogenesis of T2D through enhanced hepatic glucose output, displayed significant inflammation compared with wild-type, *Insr*^+/−^, *TIMP3*^−/−^, and *Insr*^+/−^/*Tace*^+/−^ mice. Specifically, *Insr*^+/−^/*TIMP3*^+/−^ mice display macrovesicular steatosis with severe NASH including ballooned hepatocytes, lobular and periportal inflammation and perisinusoidal fibrosis [175]. To give weight to these results, *TIMP3* expression levels were shown to be decreased in the liver of NASH patients compared to their levels in healthy controls [249], in agreement with decreased expression of *TIMP3* in the liver of mice submitted to a choline-deficient L-amino acid defined (mouse model of NASH) or western diet, in which increased oxidative stress and induction of TACE activity and fibrosis factors were observed [249]. Consistent with a protective role of TIMP3 in NASH, adenovirus-mediated injection of *TIMP3* results in decreased activity of TACE and of profibrotic factors [249]. Interestingly, exposure of stellate cells to advanced glycated end products (AGEs) was shown to decrease expression of *TIMP3* and *Sirt1* (sirtuin 1) and increase *Tace* expression through a pathway involving nicotinamide adenine dinucleotide phosphate reduced oxidase 2 (NOX2) [249]. Altogether, these results are clearly in favor of a protective role of TIMP3 from NASH development.

### 5.4. TIMP4

In contrast to TIMP3, TIMP4 deficiency is protective against HFD-induced hepatic steatosis as demonstrated by the robust decrease in liver expression of the rate-limiting enzyme stearyl CoA desaturase 1 (*Scd1*) in *TIMP4*^−/−^ mice compared with WT under a HFD [184]. SCD1 is involved in the conversion of saturated fatty acids (SFA) to monounsaturated fatty acid (MUFA) synthesis and one of the key enzymes in lipogenesis/steatosis and hepatic insulin resistance development during a HFD [250,251].

## 6. Concluding Remarks

MMPs and TIMPs have a wide range of action on several key organs of metabolic functions (Figure 3).

The study of the role of MMPs and TIMPs is difficult and involves human observations, animal models and in vitro experiments. Human studies are crucial, but usually do not offer mechanistic understanding. In vitro studies are difficult to extrapolate because they do not replicate the full complexity of the extracellular matrix and cell-cell interactions. The use of more physiological three-dimensional matrix or organoids models may be promising in unraveling the complex roles of MMP/TIMP in metabolism.

In vivo experiments via genetic manipulations are precious to understand the full spectrum of MMPs and TIMPs actions (Table 1). MMP-KO mouse lines survive to birth and often display subtle phenotypes, which highlights the high redundancy of the MMP/TIMP system in vivo, via overlapping of substrates and compensation mechanisms [2].

The results of these different experiments do not always coincide, leading to apparently contradictory data, for which several explanations can be proposed. First, despite remarkable homology between the murine and human MMP and TIMP genes, there are some interspecies differences in their functions. There are also important pitfalls when using KO mouse studies, as described by Lijnen [252]: (i) Different genetic background, or the fact that KO animals and their WT controls may not be true littermates, may explain part of the differences in study results; (ii) Data from KO mouse models should be confirmed by rescue experiments; (iii) There is fundamental physiological difference between whole body KO and tissue-specific gene inactivation, multi-tissular actions may act as a confounding factor; (iv) Gene KO may allow expression of a physiologically active, truncated form of the protein of interest. MMP inhibition experiments also face confounding factors, such as cross-inhibition of several MMPs, MMP auto-regulation, enzymatic redundancy or compensation. More efforts are required to solve these issues, and the study of the roles of MMPs and their inhibitors in metabolic functions needs to be pursued to fully unravel their role in metabolic diseases.

## 7. Highlights and Perspectives


Role of MMPs/TIMPs in adipogenesis via ECM remodeling. It is well established that MMPs and TIMPs are key factors, both positive and negative, of adipogenesis, through their ability to remodel ECM, which allows adipocytes differentiation and hypertrophy.ECM-independent roles of MMPs. Beyond their direct action on the ECM, some MMPs can influence the endocrine pathway, for instance MMP2 inducing leptin resistance by cleaving the leptin receptor.Targeting MMP/TIMP for obesity/diabetes. The development of inhibitors targeting a number of MMPs or their inhibitors when their overexpression is causally implicated in obesity or its complications may prove efficient in the fight against these diseases. Alternatively, targeting some MMPs or their inhibitors to increase their action when there are deficient may also prove efficient when necessary.Use of MMPs/TIMPs as biomarkers of metabolic diseases. Several clinical studies suggest that MMP/TIMP may serve as biomarkers of metabolic diseases and their complications, such as diabetic decompensation, diabetic nephropathy or fatty liver disease.Targeting MMP/TIMP for myopathies. Some MMPs and their inhibitors are involved in myogenesis and/or repair following injury or disease. Therefore, a better comprehension of the molecular mechanisms and signaling pathways implicated in these processes may help advance our quest for a therapy against muscle dysfunction in metabolic diseases and in ageing.


## 8. Search Strategy and Selection Criteria

The literature search was conducted using electronic databases such as PubMed, Web of Science, and Scopus. The search terms included a combination of relevant keywords such as ‘matrix metalloproteinases’, ‘tissue inhibitors of metalloproteinases’, ‘metabolism’, ‘adipose tissue’, ‘obesity’, ‘diabetes’, ‘insulin resistance’, ‘metabolic syndrome’, ‘muscle’, ‘pancreas’, ‘liver’, ‘fatty liver disease’ and other related keywords. The search was limited to articles published in English from the past 20 years. Additionally, the reference lists of selected articles were manually screened to identify any additional relevant studies or landmark studies.

## Figures and Tables

**Figure 1 ijms-24-10649-f001:**
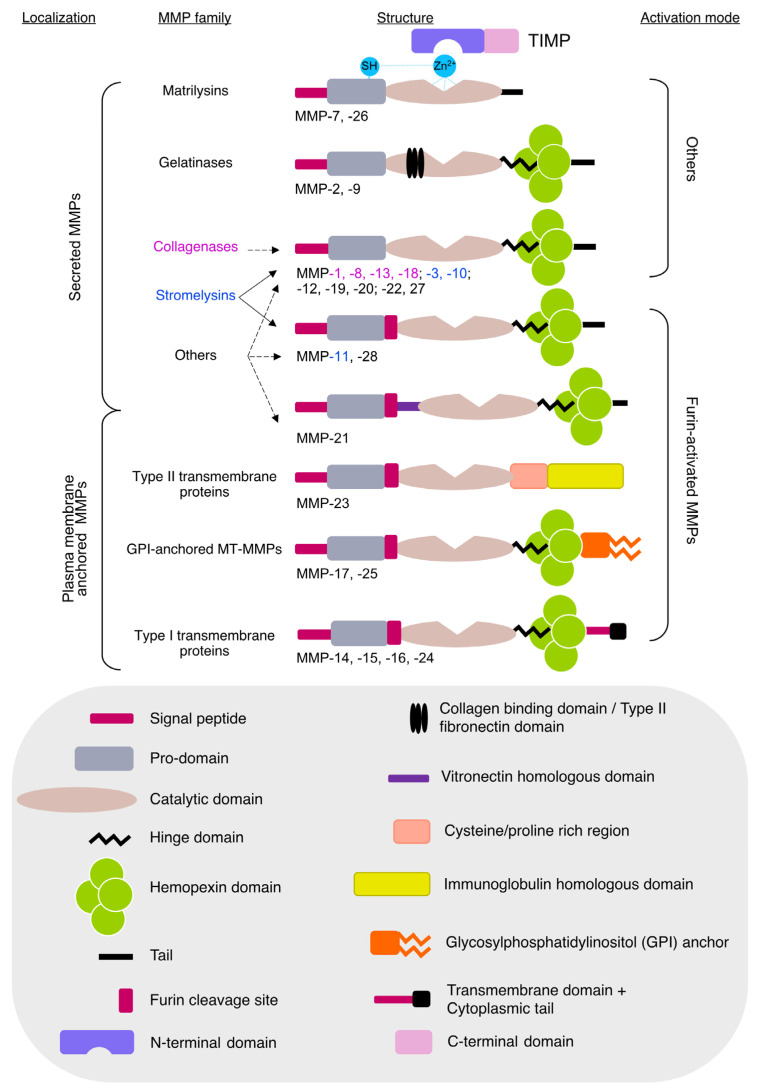
Schematic representation of MMPs and TIMPs’ structural domains. MMPs are classified based on their sequence homology but also on their cellular localization. For instance, matrilysins, gelatinases, stromelysins, and collagenases are secreted MMPs whereas membrane type MMPs (MT-MMPs) and GPI-anchored proteinases are bound to the cell membrane. MMPs can be classified based on their ability to be activated by furin. The different domains of MMPs and TIMPs are indicated in the gray box.

**Figure 2 ijms-24-10649-f002:**
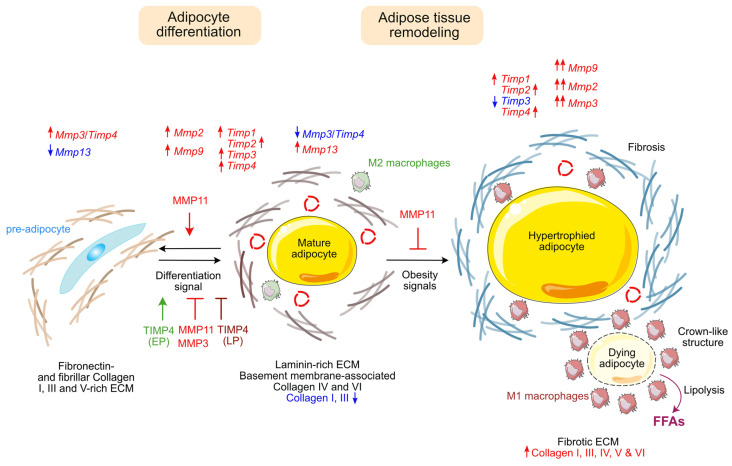
MMPs and TIMPs during adipocyte differentiation and obesity-mediated adipose tissue remodeling. In the preadipocyte state, cells express low *MMP13* levels whilst the MMP3/TIMP4 ratio is high, preventing adipocyte differentiation. During early stages of adipocyte differentiation activation, *MMP2* and *MMP9* expression levels are increased, whereas in late stages of differentiation expression of MMP inhibitors *TIMP1-4* is increased. MMP3 and MMP11 are negative regulators of adipocyte differentiation, and MMP11 favors dedifferentiation of mature adipocytes into preadipocyte-like cells with low lipid content. Mature adipocytes express high levels of *MMP13*, with a low MMP3/TIMP4 ratio. Under normal conditions, the extracellular matrix of mature white adipose tissue is vascularized and rich in laminin and expresses increased basement membrane-associated collagen IV during adipogenesis followed by its rearrangement and downregulation of collagen I and III levels. M2-type macrophages are resident cells of normal adipose tissue. When exposed to obesity signals, hypertrophied adipose tissue express high levels of *MMP2*, *MMP9* and *MMP3* whereas expression levels of *TIMP1*, *-2* and *-4* are increased and *TIMP3* decreased. The ECM of hypertrophied WAT is fibrotic and expresses increased levels of collagen I, III, IV, V and VI. Under these conditions, the polarity of macrophages is changed to the M1 pro-inflammatory type. EP: early phase differentiation, LP: late phase differentiation. Red lettering indicates an increase in local expression, blue indicates a decrease in local expression.

**Figure 3 ijms-24-10649-f003:**
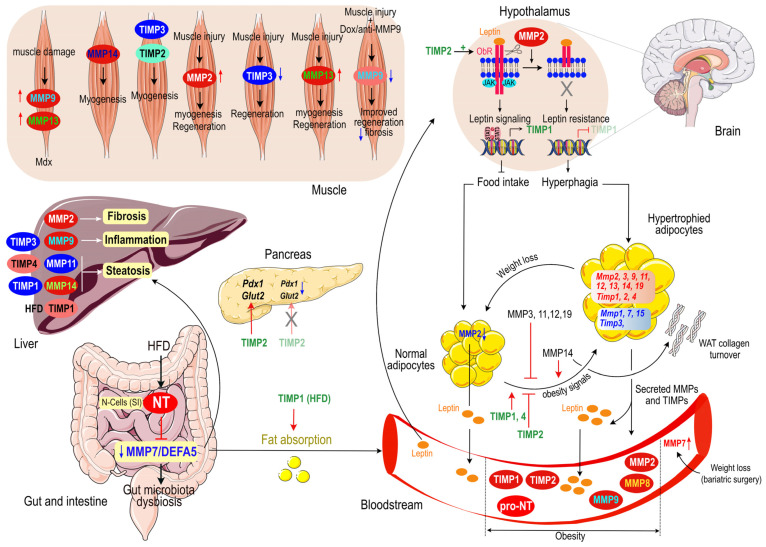
Metabolic actions of MMPs and their inhibitors. The expression of several MMPs and their inhibitors is changed in the adipose tissue in response to obesity signals, contributing to tissue remodeling. Upregulated genes are mentioned in red in the hypertrophied adipocytes, and downregulated genes are in blue. MMP3, -11, -12 and -19 as well as TIMP2 protect from diet induced obesity, whereas TIMP1 and TIMP4 are positive regulators of adipogenesis. *TIMP2* regulates positively key genes in the pancreas (*Pdx1* and *Glut2*) and its absence in the pancreas results in β-cell dysfunction and insulin resistance. However, TIMP1 and TIMP2 are negative regulators of food intake and seem to mediate leptin action in the hypothalamus. In addition, TIMP1 and TIMP2 concentrations are increased in the circulation in obese patients. Obesity results in increased circulating levels of MMP2, MMP8 and MMP9, whereas bariatric surgery-induced weight loss in obese patients is accompanied by increased circulating MMP7 levels. Increased MMP2 in obesity inhibits leptin signaling in the hypothalamus through cleavage of its receptor ObRb resulting in a state of leptin resistance and promotes hyperphagia. In the liver, increased *MMP2* expression promotes fibrosis and enhanced liver *MMP9* is proinflammatory. Increased MMP14 and TIMP4 levels; and MMP11 and TIMP3 inactivation in the liver promote liver steatosis. Moreover, a high fat diet (HFD) induces an increase in *TIMP1* expression in the liver. In the gut, a HFD induces increased neurotensin (NT) expression which inhibits the MMP7/α-defensin axis, which results in gut microbiota dysbiosis and increased intestinal fat absorption. The latter is also enhanced by HFD-mediated increase in *TIMP1* in the gut. In the muscle, MMPs play a role in myogenesis and muscle regeneration following injury or damage. MMP14 and TIMP2 are positive regulators of myogenesis and decreased *TIMP3* expression is necessary for proper myogenesis, MMP2 and MMP13 expression are positive regulators of myogenesis and muscle regeneration after muscle injury, whereas decreased TIMP3 is necessary for muscle regeneration following injury. In *Mdx* mice, muscle damage is accompanied by increased expression of *MMP9* and *MMP13*, whereas MMP9 neutralization following injury improves muscle regeneration and decreases fibrosis. Upregulated genes are indicated in red ellipses and downregulated or inactivated ones in blue ellipses.

**Table 1 ijms-24-10649-t001:** Metabolic phenotypes of MMP- and TIMP-genetically modified mice.

Genotype	Phenotype	Ref.
** *Gelatinase family* **		
*MMP2* ^−/−^	- Lower body weight - Protection against obesity in HFD - Reduced food intake - Adipocyte hypotrophy	[26]
*MMP9* ^−/−^	No difference in body weight or fat tissue composition	[40]
** *Stromelysin family* **		
*MMP3* ^−/−^	- Increased food intake during HFD - Increased body weight - Adipose tissue hypertrophy - Increased adipose tissue vascularity	[57]
*MMP11* ^−/−^	- Higher body weight (HFD and normal diet) - Metabolic syndrome phenotype	[74,76]
*MMP11^Tg^* (overexpression)	- Lower body weight - Enhanced glucose tolerance and insulin sensitivity - Mitochondrial defect, enhanced oxidative stress (in normal cells and tumor cells)	[76,78]
** *Collagenase family* **		
*MMP8* ^−/−^	- Increased body weight (only male, normal diet) - Lower serum triglyceride levels and larger HDL particles, serum cholesterol (MMP8 degrades apolipo-protein A-I)	[196]
*MMP12*^−/−^ MMP12 inhibitor	- Decreased markers of adipose tissue inflammation (crown-like structures, Nos2 expression) - Increased fat mass during HFD - Decreased FBG levels, improved glucose tolerance in mice fed a HFHS diet - Decreased FBG and improved glucose tolerance in WT mice treated with MMP12 inhibitor in a microbiota-dependent manner	[113,116] [117] [117]
** *Others* **		
*MMP19* ^−/−^	- Higher body weight upon nutrient challenge, hypertrophic adipose tissue (diet-induced obesity) - Decreased tumor susceptibility.	[139]
** *Membrane-type matrix metalloproteinases family* **		
*MMP14* ^−/−^	- Cachexia, premature death - Major metabolic alterations (decreased glycogen, lipid, circulating triglycerides and glucose) - Increased autophagy	[145]
*MMP14*^+/−^ (haploinsufficiency)	Inability to gain weight when subjected to HFD (decrease of Collagen I turnover)	[146]
***Tissue inhibitors of MMP (TIMPs*) **		
*TIMP1* ^−/−^	Tissue-specific antagonist action - *Adipose tissue action:* decreased fat mass compared to control on HFD and protection against glucose intolerance, hepatic steatosis - *Hypothalamic action:* hyperphagia on standard diet, with increased body weight.	[157,158,159]
*TIMP1^Tg^* (overexpression)	Increased adipogenesis in post-lactation breast.	[59]
*TIMP2* ^−/−^	- *Adipose tissue action*: Obesity, increased adipose tissue inflammation and insulin resistance (male only) during HFD - *Hypothalamic action*: Hyperphagia and leptin resistance.	[170,171]
Double heterozygote *TIMP3*^+/−^/*Insr*^+/−^	Overtly hyperinsulinemic and hyperglycemic, adipose tissue hypertrophy and inflammation, compared to single heterozygote mice.	[175]
Macrophage-specific *TIMP3* overexpression	Protection from HFD-induced metaflammation, insulin resistance and NASH	[176]
*TIMP4* ^−/−^	- Reduced lipid absorption (enterocyte) and lower body fat, reduced hypertrophy and fibrosis of adipose tissue - Not protected from diet-induced glucose intolerance despite reduced insulinemia - Reduced liver and skeletal muscle triglyceride accumulation and dyslipidemia.	[184]

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
