# Peer review of "Roles of Matrix Metalloproteinases and Their Natural Inhibitors in Metabolism: Insights into Health and Disease"

_ijms, 2023, doi:10.3390/ijms241310649_

Round 1

Reviewer 1 Report

This is a very thorough review on Matrix metalloproteinases (MMPs) and gives an accurate overview on the existing research with a more detailed review on the effect of MMPs to metabolic disorders like obesity, diabetes etc., as they appear to have a physiological role and the authors highlight the consequences of impaired or exacerbated MMPs actions.

Overall is a well written, easy to read review. It is quite interesting and informative as it is summarising the physiological roles of MMPs and at the same time accomplishes to accurately describe and briefly discuss the existing scientific data on each separate subject presented. The graphs and the tables provide a very well- thought summary presentation of this extensive review and are quite helpful to the reader. This review can be easily used as a concise guide to the role and function of MMPs in the metabolic disorders up to date.

There are some minor corrections the authors should notice.

The research evidence discussed are from published scientific studies but there isn’t a reference on how they were retrieved. Thus, important information is missing eg up to what year the search for the discussed papers was performed. The authors may consider to give a brief description of the methods used to identify the relevant papers. 

On specific lines some minor mistakes need attention as listed below:

Line 107 On the one hand.. please correct to: On one hand

151-157 lack of refs and some brief commenting on the others not just go and look at …

Line 216 ref of the recent studies

Line 1432 ref?

Line 1714  “…. and epigenetic modifications. “ ref is missing

Some proofreading  and editing are needed, due to the extent of the document some references are missing and some expressions  are overused eg “on the note” 

Reviewer 2 Report

Matrix metalloproteinases (MMPs) constitute a super-family of endopeptidase enzymes that degrade extracellular matrix allowing tissue remodeling during development and in the postnatal life. Their activity is regulated by natural inhibitors called TIMPs (Tissue Inhibitors of MetalloProteinases). MMPs are involved in a wide range of biological processes, both in normal physiological conditions and pathological states.

Research on MMPs has increased particularly in the last two decades and only recently their role in metabolism

has been subject of interest.

S. Molière and collaborators summarize the current knowledge regarding the metabolic roles of MMPs and TIMPs in metabolic functions and pathological conditions, focusing on locally-active MMPs and TIMPs in adipose tissue, muscle, pancreas and liver, as well as to distant actions of these proteins.

The authors have done a tremendous job critically bringing together numerous recent data on the subject, also underlining the non-concordant aspects that need clarification, insights or further study. Cited references are mostly recent and relevant publications. No similar review has been published recently; for this reason the work is extremely useful for the scientific coomunity. References to gender differences are noteworthy.

The review is well organized and comprehensively described. The data reported in the review are very numerous and structured. For this reason, the figures that summarize this extremely complex picture are very apprpriate, for example Figures 2 and 3 and Table 1.

The manuscript is accepted after minor revision.

Main concerns/comments:

Are there data on other animal models that can supplement current knowledge? if so, it would be helpful to add them.

Minor/Specific concerns: 

1.     Figure 1 legend — It would be helpful to add an explanation of the color-coding that indicates the family names (why are they shown in different colors?).

2.     Standardize the text: at paragraphs referring to the different families of MMPs in some cases the members considered are given in parentheses (i.e. line 835 Matrilysins (MMP7, MMP26)), in others they are not given (line 148, 383, 648, 769, 943…).

3.     Line 556 — “A… report reported..” just check the repetition.

4.     Line 692— Check inaccuracies in text formatting (spacing).

5.     Line 1066-7— mice is repeated.

6.     Line 1106— Check inaccuracies in text formatting (spacing).

7.     Lines 1295-1306— It is unclear why the text is formatted differently.

8.     Line 1292 “in vivo” is not written in italics

9.     Line 1412— Check the sentence ending with "with".

10.  Line 1458 “in vivo” is not written in italics

11.  Figure 2 legend— It might be helpful to add the meaning of the red and green lettering.

12.  Lines 1897-1914— It is unclear why the text is formatted differently.

13.  References 3 and 16 are the same.

Reviewer 3 Report

The submitted manuscript entitled “Metabolic functions of matrix metalloproteinases and their inhibitors in health and disease” focuses on discussion of physiological roles of MMPs and TIMPs and their involvement in the development of metabolic disorders such as obesity, fatty liver disease, and type 2 diabetes. The manuscript contains 237 references and 41 references published last 5 years. 3 Figures and one Table are presented to illustrate the results obtained. However, there are important concerns and recommendations to improve the quality of the manuscript. There are as follows:

1.     The manuscript is both poorly organized and very large. I think that it would be better to divide it into 2-3 separate articles depending on MMPs and their inhibitors, or their function in norm and pathology etc.

2.     The manuscript contains many unclear terms and phrases. For example, “metabolic functions”, “metabolic inflammation” etc. Since the term “metabolic” means “related to metabolism”, and metabolism is a set of chemical reactions and conversions of compounds and energy, processes on molecular level, the term “metabolic functions may be applied to some tissue or organ, but not molecules or processes.

3.     Before discussing MMPs in disease and influence of their inhibitors, the authors are recommended to discuss functions of MMPs in normal tissues including their role in ECM functioning.

4.     It is also recommended to give more correct definition of MMPs in the Introduction section instead of just mentioning: “ion-dependent endopeptidase”, because it is important that MMPs are calcium-dependent and zinc-containing endopeptidase.

5.     As for the manuscript content, it mostly focuses on adipose tissue function and pathology, adipocyte differentiation, obesity-associated metabolic dysfunctions. However, this is not emphasized in the Abstract and Introduction sections.

6.     Authors preferably discussed tissue inhibitors of MMPs. However, there are other types of inhibitors such as pharmacological inhibitors, which were not discussed in the manuscript.

7.     There is no clear information in the manuscript regarding possible interactions between MMPs and TIMPs at molecular level. There is only a schematic representation of MMPs and TIMPs structural domains.

8.      English language style and grammar: mistakes like stomelysins, proteainases (Fig.1 legend), inceeased (line 1401), and absence of commas (for example, line 81: in this review we will highlight….).

Some editing is required

Round 2

Reviewer 3 Report

The authors have properly addressed all my concerns and/or have provided adequate explanations

No comments